# Maintenance of mitochondrial integrity in midbrain dopaminergic neurons governed by a conserved developmental transcription factor

Federico Miozzo [1,5], Eva P. Valencia-Alarcón [1], Luca Stickley [1], Michaëla Majcin Dorcikova [1], Francesco Petrelli[2], Damla Tas[1,6], Nicolas Loncle[1,7], Irina Nikonenko[3], Peter Bou Dib [4] & Emi Nagoshi [1✉]

Progressive degeneration of dopaminergic (DA) neurons in the substantia nigra is a hallmark of Parkinson's disease (PD). Dysregulation of developmental transcription factors is implicated in dopaminergic neurodegeneration, but the underlying molecular mechanisms remain largely unknown. *Drosophila Fer2* is a prime example of a developmental transcription factor required for the birth and maintenance of midbrain DA neurons. Using an approach combining ChIP-seq, RNA-seq, and genetic epistasis experiments with PD-linked genes, here we demonstrate that *Fer2* controls a transcriptional network to maintain mitochondrial structure and function, and thus confers dopaminergic neuroprotection against genetic and oxidative insults. We further show that conditional ablation of *Nato3*, a mouse homolog of *Fer2*, in differentiated DA neurons causes mitochondrial abnormalities and locomotor impairments in aged mice. Our results reveal the essential and conserved role of *Fer2* homologs in the mitochondrial maintenance of midbrain DA neurons, opening new perspectives for modeling and treating PD.

[1] Department of Genetics and Evolution and Institute of Genetics and Genomics of Geneva (iGE3), University of Geneva, CH-1211 Geneva 4, Switzerland. [2] Department of Cell Biology, University of Geneva, Geneva, Switzerland. [3] Department of Basic Neurosciences and the Center for Neuroscience, CMU, University of Geneva, CH-1211 Geneva 4, Switzerland. [4] Institute of Cell Biology, University of Bern, CH-3012 Bern, Switzerland. [5] Present address: Neuroscience Institute - CNR (IN-CNR), Milan, Italy. [6] Present address: The Janssen Pharmaceutical Companies of Johnson & Johnson, Bern, Switzerland. [7] Present address: Puma Biotechnology, Inc., Berkeley, CA, USA. ✉email: Emi.Nagoshi@unige.ch

Midbrain dopaminergic (mDA) neurons play a central role in controlling key brain functions, including voluntary movement, associative learning, motivation and cognition[1]. The progressive and selective degeneration of mDA neurons in the substantia nigra (SN) reduces nigrostriatal DA transmission, leading to the motor symptoms of Parkinson's disease (PD). Although PD was first described over 200 years ago and is the most prevalent neurodegenerative movement disorder, the exact pathologic mechanisms remain unclear. Treatment options are limited to symptomatic relief with DA replacement therapy[2,3]. Thus, there is an urgent need to identify targets for disease-modifying therapies that reverse or hinder neurodegeneration in PD.

PD is a multifactorial disorder caused by a combination of genetic and environmental factors[4]. Increasing evidence indicates that PD risk factors lead to common pathological processes, including excessive production of reactive oxygen species (ROS), axonal pathology, neuroinflammation, defects in the ubiquitin-proteasome system, and mitochondrial dysfunction[2,5]. In particular, mitochondrial impairment is integral to most other pathological cellular processes and thought to play a central role in dopaminergic neurodegeneration. Complex I activity is reduced in the brains of PD patients, and mitochondrial toxins such as 1-methyl-4-phenyl-1,2,3,6-tetrahydropyridine (MPTP) and rotenone potently induce SN degeneration. The findings that multiple genes linked to familial PD, including *PINK1*, *Parkin*, *LRRK2*, *SNCA* and *DJ-1*, directly or indirectly affect mitochondrial physiology further support this notion[6,7].

Although several genes linked to rare monogenic forms of PD have been identified, genetic susceptibility factors for sporadic PD, which accounts for ~90% of cases, remain largely unknown. In this regard, it is noteworthy that dysregulation of transcription factors required for mDA development is implicated in PD. Numerous transcription factors required for various aspects of mDA neurons development, including *En1*, *En2*, *Lmx1a*, *Lmx1b*, *Foxa1*, *Foxa2*, *Nurr1*, *Pitx3* and *Otx2*, remain expressed in the adult brain and are crucial for the survival and function of differentiated mDA neurons in mice[8–16]. Genome-wide association studies (GWAS) have shown that genetic variants of developmental transcription factors are highly represented in sporadic PD patients[17,18]. Moreover, overexpression of *Nurr1*, *Foxa2*, *En1* or *Otx2* prevents the loss of DA neurons in murine PD models[19–21]. Therefore, a better comprehension of the genetic networks controlled by developmental transcription factors in differentiated mDA neurons may advance our molecular understanding of neurodegeneration in PD and open new therapeutic options.

*Drosophila* offers a powerful model system to investigate molecular mechanisms of PD pathogenesis in vivo, due to the abundance of genetic tools, the conservation of most PD-related pathways and familial PD-linked genes, and its relatively simple nervous system[22,23]. The adult fly midbrain contains approximately a dozen clusters of DA neurons projecting to different areas[24]. Although the anatomical arrangements of DA neurons in *Drosophila* and vertebrates differ considerably, recent works have highlighted the functional homology of some of the dopaminergic circuits[25]. The protocerebral anterior medial (PAM) cluster is the largest subgroup containing ~80% of all *Drosophila* brain DA neurons. PAM neurons are highly heterogeneous in their function and projection patterns, modulating various behavior including associative learning, sleep, and locomotion[26,27]. Remarkably, some of the PAM neurons are required for the startle-induced climbing, a locomotor behavior found to be defective in multiple PD models[27–29], rendering the PAM cluster partially analogous to the mammalian SN. Dopaminergic innervation from the PPL1 and PPM3 clusters to the central complex also appear to be functionally homologous to the nigrostriatal pathway and is reported to be impaired in some PD models[25,30,31].

The notion that developmental transcription factors play critical roles in the maintenance of adult DA neurons is shared between flies and mammals, as supported by our recent studies of *Drosophila Fer2* gene[28]. *Fer2* encodes a basic helix-loop-helix (bHLH) transcription factor required for the development of DA neurons in the PAM cluster; thus, fewer PAM neurons are formed in *Fer2*[2] hypomorphic mutants. Additionally, *Fer2*[2] mutant flies display a progressive loss of PAM neurons in adulthood, locomotor impairment that can be improved by L-DOPA administration, and increased ROS levels in the brain. PAM neurons deficient for *Fer2* accumulate abnormal mitochondria and show autophagy dysfunction. *Fer2* expression persists into adulthood in PAM neurons and increases in response to oxidative stress. Importantly, the maintenance of mitochondrial integrity and the survival of PAM neurons in aged flies require the presence of *Fer2* within PAM neurons in adulthood[28,29]. Taken together, these lines of evidence establish the dual role of *Fer2* in development and life-long maintenance of DA neurons in the PAM cluster, and suggest that molecular studies of *Fer2* function will shed light on the mechanisms of dopaminergic neurodegeneration relevant to PD.

In the present study, we ask how *Fer2* maintains DA neurons throughout adulthood and whether its function is conserved in mammals. We address these questions by analyzing functional interaction between *Fer2* and familial PD-linked genes, by identifying *Fer2* target genes and by generating mice with a conditional deletion of *Nato3*, a murine homolog of *Fer2*, in differentiated DA neurons. This combined approach demonstrates that *Fer2* is a master regulator of a transcriptional network controlling mitochondrial integrity and function, and thereby confers dopaminergic neuroprotection in multiple genetic and toxin models of PD. *Nato3* is known to be required for the genesis of mDA neurons[32–34] but its post-developmental role has never been studied. We demonstrate here that selective ablation of *Nato3* in differentiated DA neurons leads to age-dependent motor impairment and a loss of mitochondrial integrity in mDA neurons. Taken together, our results indicate that *Fer2* and *Nato3* share an essential role in maintaining mitochondrial health and dopaminergic function, offering new opportunities to study the selective vulnerability of mDA neurons and to develop therapeutic interventions for PD.

## Results

***Fer2* overexpression protects PAM dopaminergic neurons from genetic and oxidative insults.** To explore the genetic mechanisms underlying the role of *Fer2* in the maintenance of DA neurons, we investigated whether *Fer2* loss or gain of function affects the survival of DA neurons in genetic models of PD. Mutations in the *leucine-rich repeat kinase 2* (*LRRK2*) gene are the most common cause of familial PD and also predispose carriers to sporadic PD[35]. Because pathogenic *LRRK2* variants are considered to be gain-of-function mutants, most fly models of *LRRK2*-linked PD employ overexpression of wild-type or mutant forms of *LRRK2* or its fly homolog. We thus overexpressed wild-type or mutant *Drosophila Lrrk* via the GAL4/UAS system using PAM neuron-specific drivers, *Fer2-GAL4* or *R58E02-GAL4*[26,28]. The number of surviving DA neurons was assessed by immunostaining with anti-tyrosine hydroxylase (TH) antibodies.

Forced expression of either wild-type or *Lrrk I1915T* mutant, which is homologous to the human pathogenic *LRRK2 I2020T* variant[35,36], reduced the number of TH-positive neurons in the PAM cluster by ~10% in 1-day-old flies (Supplementary Fig. 1a). Thereafter TH-positive cell counts further decreased in an age-

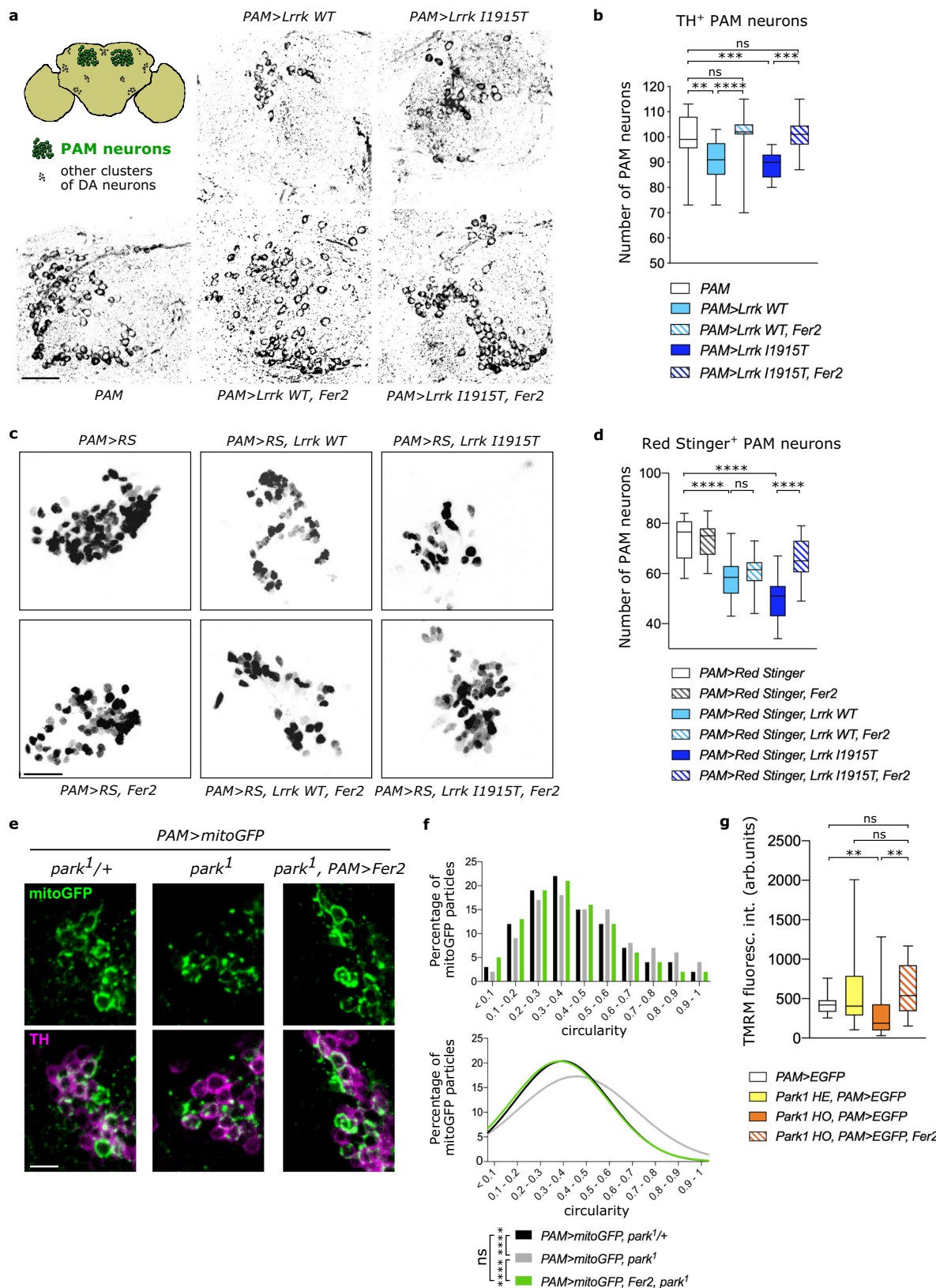

dependent manner (Supplementary Fig. 1a and Fig. 1a, b). To distinguish the reduction in TH expression and the genuine loss of DA neurons, we also visualized PAM neurons by expressing the nuclear RedStinger red fluorescent protein together with *Lrrk*. The number of RedStinger-positive cells was reduced in the flies expressing wild-type or mutant *Lrrk* in PAM neurons (Fig. 1c, d).

Therefore, *Lrrk* gain of function causes not only developmental loss but also age-dependent loss of PAM neurons. Remarkably, overexpression of *Fer2* in PAM neurons prevented the reduction of TH-positive cell counts induced by the expression of wild-type or mutant *Lrrk* (Fig. 1a, b). The analysis of cell counts using RedStinger found that PAM neuron loss caused by the mutant

**Fig. 1 _Fer2_ overexpression prevents the loss of PAM dopaminergic neurons and mitochondrial dysfunction in genetic models of PD. a, b** _Fer2_ overexpression prevented the loss of PAM neurons induced by targeted expression of wild-type (_WT_) or pathogenic mutant _Lrrk_ in PAM neurons. **a** Representative images of PAM neurons at day 14 detected by anti-TH antibodies (green). _PAM_, PAM neuron-specific _R58E02-GAL4_ driver. _PAM > X_ indicates that _UAS-transgene X_ is driven by _R58E02-GAL4_. Scale bar, 20 μm. Top left, a schematic of DA neurons in the PAM (green) and other clusters (black). **b** Quantification of the number of TH[+] PAM neurons per hemisphere from (**a**). $n = 16$ hemispheres per group. **$p < 0.01$, ***$p < 0.001$, ****$p < 0.0001$. _PAM_ vs. _PAM > Lrrk WT_, $p = 0.0026$. _PAM_ vs. _PAM > Lrrk I1915T_, $p = 0.0002$. _PAM > Lrrk I1915T_ vs. _PAM > Lrrk I1915T, Fer2_, $p = 0.0001$. ns, not significant. One-way ANOVA followed by a Tukey's test for multiple group comparison. **c** Representative images of PAM neurons at day 14 labeled by _PAM > Red Stinger_. Scale bar, 20 μm. **d** Quantification of the number of Red Stinger[+] PAM neurons per hemisphere from (**c**). $n = 16$ hemispheres per group. ****$p < 0.0001$. ns, not significant. One-way ANOVA followed by a Tukey's test for multiple group comparison. **e** The mitochondrial network in PAM neurons was visualized using mitoGFP. Scale bar, 5 μm. **f** Top, histograms showing the circularity distribution of GFP[+] particles measured from (**e**). Bottom, the Gaussian curve fit to the circularity histogram. Circularity ranges from 0 (reticular mitochondria) to 1 (round mitochondria). ****$p < 0.0001$ using Kruskal–Wallis test followed by Dunn's test for multiple group comparison. 500 mitochondrial particles were analyzed from 16 hemispheres per genotype. **g** Quantification of TMRM fluorescence intensity in GFP-labeled PAM neurons in 14-days-old flies. $n = 18$ hemispheres per group. HE, heterzogous. HO, homozygous. **$p < 0.01$. _PAM > EGFP_ vs. _Park1 HO, PAM > EGFP_, $p = 0.0063$. _Park1 HO, PAM > EGFP_ vs. _Park1 HO, PAM > EGFP, Fer2_, $p = 0.0016$. ns, not significant. One-way ANOVA followed by a Tukey's test for multiple group comparison. Box boundaries in (**b**), (**d**) and (**g**) are the 25th and 75th percentiles, the horizontal line across the box is the median, and the whiskers indicate the minimum and maximum values.

_Lrrk_, but not the WT _Lrrk_, was rescued with _Fer2_ overexpression at least partially (Fig. 1c, d). Forced expression of _Lrrk_ in _Fer2²_ heterozygous background resulted in PAM neuron loss to a similar extent as in wild-type background (Supplementary Fig. 1b). _Lrrk_ overexpression in _Fer2²_ homozygous mutants additively enhanced PAM neuron loss compared with either of the single genetic perturbations (Supplementary Fig. 1b). These results suggest that _Fer2_ and _Lrrk_ act in independent pathways but _Fer2_ overexpression can attenuate _Lrrk_-induced neuronal damage.

Mutations in the _PRKN_ gene, which encodes the E3 ubiquitin ligase Parkin, are linked to autosomal recessive juvenile PD[37]. _Drosophila parkin_ loss-of-function mutant _park¹_ exhibits PD-like phenotypes, including locomotor deficits, mitochondrial impairments and loss or shrinkage of a few subclasses of DA neurons[38,39]. Since the effect of _parkin_ loss of function on the integrity of DA neurons in the PAM cluster has not been evaluated in previous studies, we visualized PAM neurons using anti-TH immunostaining and targeted expression of RedStinger in _park¹_ mutants. An age-dependent reduction in the number of TH-positive PAM neurons was observed in _park¹_ homozygotes (Supplementary Fig. 1c). However, the number of cells expressing RedStinger with the PAM neuron driver was not significantly different between the heterozygous and homozygous _park¹_ mutants (Supplementary Fig. 1d). Therefore, _park¹_ mutation reduces TH expression in PAM neurons, which likely leads to functional impairments, although it does not cause cell loss.

Parkin mutations lead to severe defects in mitochondrial morphology and function in all organisms studied thus far, including flies[40–43]. We therefore examined whether _Fer2_ overexpression ameliorates mitochondrial morphology in DA neurons of _park¹_ flies by expressing mitochondria-targeted GFP (mitoGFP) in PAM neurons. In contrast to the reticular mitochondria observed in control flies, the majority of mitochondria appeared more rounded in PAM neurons of _park¹_ mutants (Fig. 1e). This observation was quantitatively confirmed by the analysis of mitochondria circularity, which showed a shift towards more globular mitochondria in _park¹_ compared with control (_park¹/+_) flies (Fig. 1f). Remarkably, the overexpression of _Fer2_ in PAM neurons restored tubular mitochondrial morphology in _park¹_ flies (Fig. 1e, f). Furthermore, to determine if _Fer2_ overexpression improves the functioning of mitochondria in _park¹_ flies, we assessed mitochondrial membrane potential in PAM neurons using the tetramethylrhodamine methyl ester (TMRM) staining. TMRM fluorescence levels were significantly reduced in homozygous _park¹_ mutants compared to _w¹¹¹⁸_, indicating the decrease in mitochondrial membrane potential as a

result of _parkin_ loss of function. Overexpression of _Fer2_ increased TMRM fluorescence to the levels equivalent to those in _w¹¹¹⁸_ and _park¹/+_ flies (Fig. 1g). These results indicate that _Fer2_ can counteract both morphological and functional impairments in mitochondria caused by loss of _parkin_.

The findings that _Fer2_ overexpression cell-autonomously prevents cellular and mitochondrial damage in the PAM cluster DA neurons in two independent familial PD models suggest that _Fer2_ gain of function might confer dopaminergic neuroprotection against a broad spectrum of genetic and chemical insults. We therefore tested if _Fer2_ overexpression can protect DA neurons from damage caused by oxidative stress, using an assay in which flies were exposed to a non-lethal dose of hydrogen peroxide ($H_2O_2$). 7-day-old flies were fed with 5% $H_2O_2$ for 24 h, and 7 days later DA neurons in the brains were examined by anti-TH immunostaining. Whereas this treatment does not immediately affect PAM neuron integrity in control flies[28,29], 7 days later it triggered the loss of ~10% of DA neurons in the PAM cluster in _w¹¹¹⁸_ flies, _Fer2²/+_ flies and flies carrying a PAM-specific driver (Fig. 2a). The PAM neuron loss was exacerbated in flies homozygous for the _Fer2²_ allele, consistent with our previous finding that _Fer2_ inactivation renders PAM neurons more vulnerable to oxidative stress[28,29]. In contrast, _Fer2_ overexpression in PAM neurons prevented $H_2O_2$-induced PAM neuron loss (Fig. 2a). These results indicate that _Fer2_ gain of function protects DA neurons in the PAM cluster from oxidative insults.

ROS production increases as flies age[44]. Interestingly, we found an age-dependent increase in FER2 protein levels in PAM neurons, parallel to the augmentation of ROS levels in the brain (Fig. 2b, e). Combined with the fact that _Fer2_ loss of function causes a progressive loss of PAM neurons, this finding suggests that FER2 is upregulated to protect PAM neurons from age-dependent increases in endogenous oxidative stress. Taken together, these results provide evidence that _Fer2_ is a potent neuroprotective transcription factor for PAM dopaminergic neurons, counteracting various genetic and oxidative insults.

**Identification of _Fer2_ direct targets by a combination of ChIP-seq and RNA-seq approaches.** To investigate how _Fer2_ elicits dopaminergic neuroprotection, we wished to identify _Fer2_ transcriptional targets. To this end, we first determined the genome-wide binding sites of FER2 using chromatin immunoprecipitation coupled to sequencing (ChIP-seq). Since FER2-specific antibodies were unattainable, ChIP-seq was performed on the heads of 14-day-old flies harboring a _Fer2::GFP::V5_ genomic transgene in _Fer2¹_ mutant background[28], using anti-V5 antibody (Fig. 3a). _Fer2::GFP::V5_ recapitulates the physiological dosage and

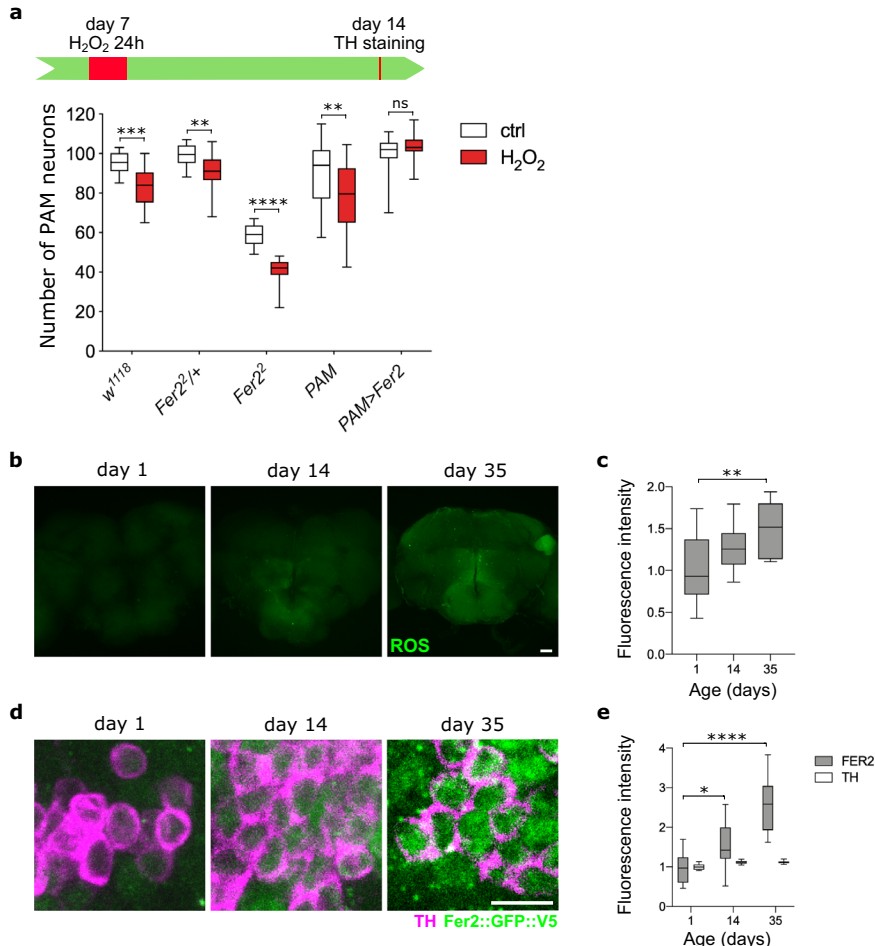

**Fig. 2 *Fer2* overexpression protects PAM dopaminergic neurons from oxidative insults. a** $w^{1118}$, $Fer2^2/+$, $Fer2^2$, R58E02-GAL4 (PAM) and PAM > *Fer2* flies were fed with $H_2O_2$ at day 7 for 24 h and the number of PAM neurons was examined at day 14. *Fer2* mutation exacerbates, whereas *Fer2* overexpression prevents, $H_2O_2$-induced PAM neuron loss. $n = 16$ hemispheres per group. Two-tailed Mann–Whitney test, **$p < 0.01$, ***$p < 0.001$, ****$p < 0.0001$. $w^{1118}$ control (ctrl) vs. $H_2O_2$, $p = 0.001$. $Fer2^2/+$ ctrl vs. $H_2O_2$, $p = 0.0076$. *PAM* ctrl vs. $H_2O_2$, $p = 0.0032$. ns, not significant. **b** ROS levels in $w^{1118}$ flies at different ages were monitored using H2DCF dye. Scale bar, 25 μm. **c** Quantification of ROS levels in the central brain from (**b**). Data are represented as fold change relative to day 1. $n = 14$ hemispheres per group. One-way ANOVA followed by a Tukey's test for multiple group comparison, **$p < 0.01$. day 1 vs. day 35, $p = 0.0048$. **d** Brains of *Fer2::GFP::V5* flies of different ages were stained with anti-GFP (green) and PAM neurons were detected by anti-TH antibodies (magenta). Scale bar, 5 μm. **e** GFP and TH fluorescence intensities in the PAM region from (**d**) were quantified. Data are represented as fold change relative to day 1. $n = 16$ hemispheres per group. One-way ANOVA followed by a Tukey's test for multiple group comparison, *$p < 0.05$, ****$p < 0.0001$. day 1 vs. day 35, $p = 0.0117$. Box boundaries in (**a**), (**c**) and (**e**) are the 25th and 75th percentiles, the horizontal line across the box is the median, and the whiskers indicate the minimum and maximum values.

expression pattern of *Fer2*, as it is expressed in PAM neurons but scarcely in other cells in the brain (Supplementary Fig. 2a), in agreement with the existing single-cell RNA-seq data[45,46]. We also found that *Fer2::GFP::V5* rescues PAM neuron loss and locomotor defects in *Fer2*[1] mutants, verifying that FER2::GFP fusion protein retains the function of FER2 (Supplementary Fig. 2b, c). We identified 269 FER2-bound regions, which were annotated to 279 putative *Fer2* target genes (Supplementary Data 1). FER2-bound sequences were highly enriched with a 12-nucleotide motif containing a canonical bHLH binding site (CANNTG), validating the specificity of the procedure (Fig. 3b).

The binding of transcription factors to chromatin does not necessarily lead to transcriptional regulation of nearby genes[47]. To determine which binding events regulate proximal gene expression, we next identified transcripts that are up- or downregulated after a transient overexpression of *Fer2* and compared these transcripts with the ChIP-seq data. The heat-inducible *hs-GAL4* was used to ubiquitously but acutely express *Fer2*, and RNA-seq was performed on fly heads collected 12 and

48 h after a 3-h heat shock. The same heat-shock treatment was applied to the flies carrying *hs-GAL4* but not *UAS-Fer2* to identify and exclude from our analysis any heat-responsive gene (Fig. 3c). These experiments identified 328 genes (12 h after heat shock) and 916 genes (48 h after heat shock) differentially expressed in the head following *Fer2* overexpression (Fig. 3c and Supplementary Data 2). Then, by comparing the head RNA-seq and *Fer2* ChIP-seq data, we found 32 genes that were present in FER2 ChIP-seq and also differentially expressed 48 h after *Fer2* overexpression (Fig. 4b). These genes are considered as potential *Fer2* transcriptional targets. The overlap was statistically highly significant (hypergeometric probability = 0.0004). In contrast, there was no statistically significant overlap between the ChIP-seq data and the genes differentially expressed 12 h after heat-shock, therefore we focused our subsequent analysis on the data set obtained at the 48 h timepoint. The ChIP-seq profiles and RNA-seq data of two *Fer2* targets, *CG8128* and *skd*, are presented as examples in Fig. 3d. Because *Fer2* expression is largely restricted to PAM neurons, the majority of FER2 ChIP signal represents

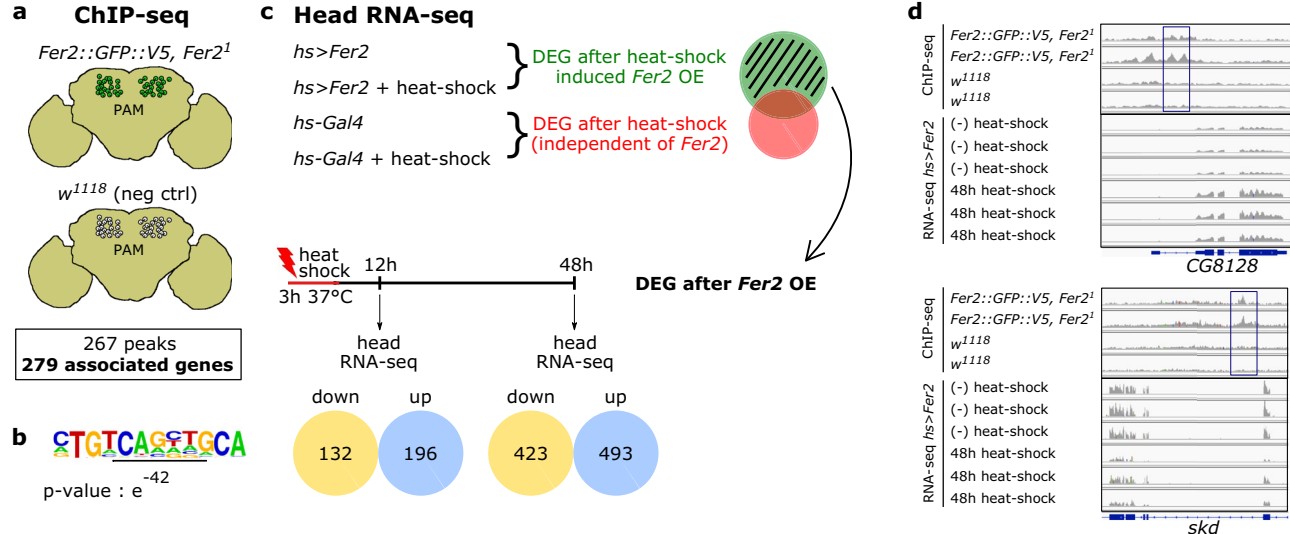

**Fig. 3 Identification of *Fer2* transcriptional targets through a combination of ChIP-seq and RNA-seq approaches. a** ChIP-seq was performed on 14-day-old *Fer2::GFP::V5, Fer2¹* flies. *w¹¹¹⁸* flies were used as a negative control. *n* = 2 biological replicates per group. **b** A sequence containing an E-box (CANNTG, underlined) was found enriched by Fisher's exact test in FER2 ChIP-seq peaks over *w¹¹¹⁸* ChIP-seq. **c** Scheme of the whole-head RNA-seq analysis to identify DEG after heat shock-driven *Fer2* transient overexpression. *n* = 3 biological replicates per group. The number of DEGs up- and downregulated after heat shock are shown. **d** Examples of ChIP-seq and RNA-seq data (48 h) for two *Fer2* direct targets, *CG8128* and *skd*. The rectangles indicate ChIP-seq peaks.

FER2 binding within PAM neurons, whereas whole-head RNA-seq following *Fer2* induction detects genes also expressed outside the PAM cluster. Therefore, to determine whether the *Fer2* targets identified above are expressed in PAM neurons, we profiled the transcriptome of isolated PAM neurons (Fig. 4a). Indeed, the majority (26 out of 32) of *Fer2* target genes identified by the ChIP-seq/head RNA-seq intersectional analysis were found to be expressed in PAM neurons (Fig. 4b), validating this subset of 26 genes as bona fide direct transcriptional targets of *Fer2* (Fig. 4c, Supplementary Data 3).

A further analysis of the 26 bona fide targets revealed that *Fer2* preferentially binds to introns of the target genes and acts as a repressor in most instances, as the majority of direct targets are downregulated after acute *Fer2* overexpression (Fig. 4d, e). A large fraction of the 26 *Fer2* direct targets are involved in the regulation of gene expression, including transcription factors (*ftz-f1*, *Eip75B*, *Eip93F*, and *Snoo*), transcriptional coregulators (*skd*), components of chromatin remodeling complexes (*Smr*, *osa*, and *sba*) and histone modifiers (*gpp*). These genes show a significant interconnectivity (*p* = 0.00006) either physically or functionally, as inferred from the STRING database analysis, suggesting that they may cooperate to regulate downstream gene expression (Fig. 4c).

**Multiple *Fer2* direct target genes contribute to the maintenance of PAM dopaminergic neurons**. Next, we assessed whether *Fer2* direct target genes have roles similar to *Fer2* by manipulating their expression levels individually in PAM neurons and monitoring neuronal survival. To mimic the effect of *Fer2* loss of function, target genes upregulated by *Fer2* were knocked-down via RNAi, and genes downregulated by *Fer2* were over-expressed in PAM neurons. PAM neurons were examined at the age of 35 days with anti-TH staining. 17 out of 26 *Fer2* direct targets were tested in this assay, since transgenic lines for RNAi or overexpression for 7 genes were unavailable or lethal when combined with the *Fer2-GAL4* driver and two other genes, *CG1578* and *Theg*, could not be efficiently downregulated with available *UAS-RNAi* lines (~30% downregulation assessed by RT-

qPCR). Of note, driving *UAS-CG4998 RNAi* with the ubiquitously expressing *tubulin-GAL4* or *actin-GAL4* was lethal, which suggests a high efficacy of the UAS-RNAi line for *CG4998*. The fly lines used in this assay are described in Supplementary Data 3 along with their references. Genetic manipulation of one third of the tested genes (6 out of 17) caused a significant loss of PAM neurons compared with the driver-only control, thus partially recapitulating the phenotype of *Fer2* loss of function (Fig. 4f and Supplementary Data 3). These results broadly confirm that the ChIP-seq and RNA-seq intersectional analysis identified a panel of *Fer2* direct targets that contribute to the role of *Fer2* in the development and maintenance of PAM neurons.

**Fer2 is a master regulator of mitochondrial health and function**. Since many *Fer2* direct targets are transcriptional regulators, *Fer2* most likely orchestrates the expression of a larger set of secondary targets to promote the survival of PAM neurons. To decipher downstream biological processes controlled by *Fer2*, we also profiled the transcriptome of isolated PAM neurons following PAM-specific, constitutive *Fer2* overexpression (Fig. 4a). We reasoned that this approach should capture the long-term consequences of *Fer2* overexpression on gene expression in PAM neurons. A comparison of PAM neuron transcriptomes of flies overexpressing *Fer2* and age-matched controls identified 132 differentially expressed genes (DEGs) (24 genes upregulated and 108 genes downregulated; *p* < 0.01). These genes were distinct from the 26 direct targets and were considered as *Fer2* secondary targets (Fig. 4a, b; Supplementary Data 4).

Gene ontology (GO) analysis of *Fer2* secondary targets revealed a marked enrichment of nuclear-encoded mitochondrial genes among up-regulated genes. Down-regulated genes are enriched with the GO terms "translational termination" and "vacuoles" (Fig. 4g). Up-regulated mitochondrial genes include a component of the electron transport chain (ETC) (*ND-15*), genes involved in the assembly or maturation of ETC complexes (*Sdhaf3*, *CG7639*, *Cchl*, *CG5037*), and the mitochondrial *Superoxide dismutase 2* (*Sod2*) (Fig. 4g and Supplementary Data 4). As a crucial player in the cellular antioxidant response, SOD2 is localized to the

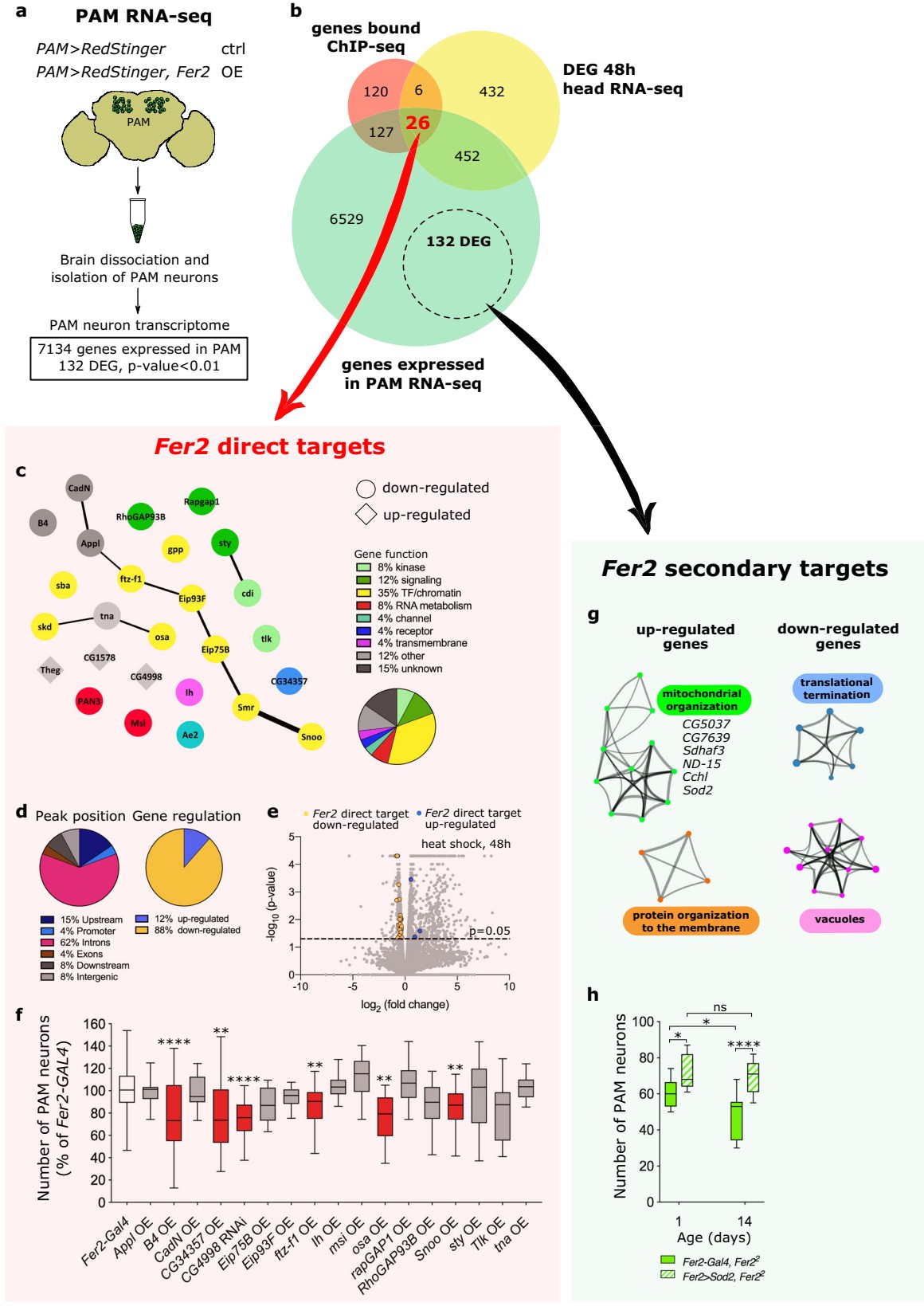

**a** PAM RNA-seq

*PAM>RedStinger* ctrl
*PAM>RedStinger, Fer2* OE

Brain dissociation and isolation of PAM neurons

PAM neuron transcriptome

7134 genes expressed in PAM
132 DEG, p-value<0.01

**b** genes bound ChIP-seq

DEG 48h head RNA-seq

genes expressed in PAM RNA-seq

132 DEG

***Fer2* direct targets**

**c**

○ down-regulated
◇ up-regulated

Gene function
8% kinase
12% signaling
35% TF/chromatin
8% RNA metabolism
4% channel
4% receptor
4% transmembrane
12% other
15% unknown

**d** Peak position | Gene regulation

15% Upstream
4% Promoter
62% Introns
4% Exons
8% Downstream
8% Intergenic

12% up-regulated
88% down-regulated

**e** ● *Fer2* direct target down-regulated ● *Fer2* direct target up-regulated

heat shock, 48h

p=0.05

**f**

***Fer2* secondary targets**

**g** up-regulated genes | down-regulated genes

mitochondrial organization
CG5037
CG7639
Sdhaf3
ND-15
Cchl
Sod2

translational termination

protein organization to the membrane

vacuoles

**h**

mitochondrial matrix and catalyzes superoxide ($O_2 \cdot^-$) to the less reactive $H_2O_2$[48]. Remarkably, *Sod2* overexpression in PAM neurons partially but significantly prevented neuronal loss in *Fer2*[2] mutant flies, which suggests that *Sod2* acts downstream of *Fer2* to contribute to dopaminergic neuroprotection (Fig. 4h). Combined with the finding that *Fer2* regulates the expression of

numerous mitochondrial genes, these results demonstrate that *Fer2* plays a prominent role in the maintenance of mitochondria.

To explore how *Fer2* direct targets control a large set of secondary targets, we searched for transcription factor binding sites that are significantly overrepresented in the promoters of DEGs in PAM RNA-seq. Seven binding motifs were found to be

**Fig. 4 Fer2 targets contribute to PAM neurons maintenance and mitochondrial health. a** Scheme of RNA-seq analysis of isolated PAM neurons. $n = 3$ biological replicates per group. **b** Intersection between genes identified by FER2 ChIP-seq, DEG identified by whole-head RNA-seq at 48 h, and genes expressed in PAM neuron RNA-seq. **c** Network visualization of the 26 *Fer2* direct targets and their interactions. Each node represents a gene and is colored based on its function. The line thickness indicates the strength of data support. **d** Position of FER2 peaks and effect of *Fer2* overexpression on *Fer2* direct targets levels. **e** Volcano plot of 48 h RNA-seq data. Changes in gene expression were tested by t test with Cuffdiff2 tool. Orange (downregulated) and blue (upregulated) dots indicate the 26 *Fer2* direct targets. **f** Quantification of PAM neurons per hemisphere detected by anti-TH immunostaining in 35-day-old flies, upon overexpression (OE) or knock-down (RNAi) of the indicated *Fer2* direct targets in PAM neurons. $n = 14$ hemispheres per group. Genotypes causing a statistically significant PAM neuron loss are highlighted in red. A second independent replicate was assessed for these genotypes to confirm the result. One-way ANOVA followed by a Dunnett's test for multiple group comparison, **$p < 0.01$, ****$p < 0.0001$. *Fer2-Gal4* vs. *CG34357* OE, $p = 0.0012$. *Fer2-Gal4* vs. *ftz-f1* OE, $p = 0.0031$. *Fer2-Gal4* vs. *osa* OE, $p = 0.0070$. *Fer2-Gal4* vs. *Snoo* OE, $p = 0.0015$. **g** Network visualization of the most enriched GO terms among up-regulated and down-regulated DEGs in PAM RNA-seq. GO terms were grouped into clusters based on their membership similarities. Each node represents a GO term and is colored based on its cluster ID. GO terms not clustering are not shown. GO terms are detailed in Supplementary Data 5. **h** Quantification of PAM neurons per hemisphere, as detected by anti-TH immunostaining. $n = 16$ hemispheres per group. One-way ANOVA followed by a Tukey's test for multiple group comparison, *$p < 0.05$, ****$p < 0.0001$. *Fer2-Gal4, Fer2²* day1 vs. *Fer2-Gal4, Fer2²* day14, $p = 0.0152$. *Fer2-Gal4, Fer2²* day1 vs. *Fer2 > Sod2, Fer2²* day1, $p = 0.0191$. Box boundaries in (**f**) and (**h**) are the 25th and 75th percentiles, the horizontal line across the box is the median, and the whiskers indicate the minimum and maximum values.

---

enriched, including those for Broad, Dichate and USP, which are expressed in PAM neurons. Interestingly, these genes are known to form complexes or functionally interact with FTZ-F1, a *Fer2* direct target[49,50] (STRING database, https://string-db.org/) (Supplementary Fig. 3a, b). Consistent with the finding that *ftz-f1* is downregulated by FER2, *ftz-f1* overexpression leads to the loss of PAM neurons (Fig. 4f). Therefore, gene regulatory networks that control PAM neuron maintenance initiated by *Fer2* may be mediated by FTZ-F1 together with these factors. Another *Fer2* direct target, SMRTR, can interact with USP[51], and may thus initiate a second pathway to control gene networks in PAM neurons (Supplementary Fig. 3a, b).

**Conditional deletion of *Nato3* in differentiated DA neurons leads to locomotor deficits in aged mice.** Since *Fer2* is required for the maintenance of mitochondrial health and integrity in DA neurons, we wondered whether the mammalian homolog of *Fer2* has a similar role. Blast searches found four mammalian bHLH transcription factors with a significant similarity to FER2, namely P48/PTF1A, NATO3 (N-TWIST, FERD3L), TCF15, and TWIST1 (Fig. 5a). Interestingly, it has been shown that *Nato3* is expressed in the progenitors of mDA neurons and is required for dopaminergic neurogenesis[32,33,52], analogously to the role of *Fer2* in the genesis of DA neurons in flies[28]. Furthermore, RNA-seq data from the Genotype-Tissue Expression (GTEx) project (the GTEx Consortium, Nat Genetics, 2013)[53] indicate that, among human tissues, *NATO3* expression is relatively restricted to the SN of the adult brain (Fig. 5b), whereas the other three *Fer2* homologs show little or no expression in the SN (Supplementary Fig. 4b–d). Based upon these evident similarities in the protein sequence, the expression in mDA neurons and the role in the development of mDA neurons, we consider *Nato3* to be a functional homolog of *Fer2*.

To investigate whether *Nato3* is involved in the maintenance of postnatal mDA neurons, we first determined its expression levels and specificity in the mouse brain. Reverse Transcription quantitative PCR (RT-qPCR) analysis of dissected brain regions displayed a significant enrichment of *Nato3* transcripts in the midbrain of 4-month-old mice (Fig. 5c). *Nato3* continues to be expressed at similar levels in the midbrain of young (1-month-old), adult (4-month-old) and aged (14-month-old) mice (Fig. 5d). In situ hybridization combined with anti-TH immunostaining showed *Nato3* mRNA expression in the DA neurons of the SN and the ventral tegmental area (VTA) of 14-month-old mice. *Nato3* expression was also detected in non-TH-positive cells in the SN and VTA (Fig. 5e, f).

Having verified the expression of *Nato3* in adult mDA neurons, we sought to conditionally delete *Nato3* in postmitotic DA neurons and investigate its consequences to dopamine-related behavior and the integrity of mDA neurons. We generated a mouse strain bearing the *loxP*-flanked (floxed) allele of *Nato3* and crossed it with the *DAT-Cre* transgenic mouse expressing the Cre recombinase under the control of the dopamine transporter gene (*Slc6a3*) regulatory sequences[54]. *DAT-Cre* expression is known to arise postnatally in differentiated midbrain DA neurons, has no deleterious effects on DA neurons, and does not alter locomotion or anxiety-related behaviors[54,55]. We obtained homozygous *Nato3*-floxed animals harboring either no copies (*Nato3^floxed*, referred to as control) or one copy of the *DAT-Cre* transgene (*Nato3^DAT-Cre*, referred to as *Nato3* cKO) (Fig. 6a). RT-qPCR analysis showed ~50% reduction of *Nato3* expression in the ventral midbrain in *Nato3^DAT-Cre* mice compared to controls (Fig. 6b). In situ hybridization revealed the reduction of the percentage of TH⁺ neurons expressing *Nato3* mRNA from ~90% to ~23% in *Nato3* cKO (Fig. 6c, d), which confirms the high efficiency of *DAT-Cre* mediated *Nato3* ablation in mDA neurons.

We analyzed control and *Nato3* cKO mice at different ages using assays for dopamine-related behaviors. In the pole test, which is a sensitive assay for motor impairment associated with reduced striatal dopamine in rodents[56], *Nato3* cKO mice took significantly longer to turn and descend the pole than control mice at 13 months, while no difference was observed in younger animals (Fig. 6e). This observation indicates an age-dependent decline in motor coordination in *Nato3* cKO mice. Spontaneous activity levels measured via voluntary wheel running over three days were similar between *Nato3* cKO and control mice until 13 months of age. However, the activity levels of *Nato3* cKO significantly declined at 16 months (Fig. 6f). In contrast, no differences were found between *Nato3* cKO and control mice in the open field test, which measures exploratory behavior when mice are placed in a novel environment[57] (Supplementary Fig. 5). These results indicate that *Nato3* ablation in postmitotic DA neurons lead to motor abnormalities in aged mice, possibly caused by an age-dependent decline in the nigrostriatal dopaminergic function.

**Nato3 is required for the maintenance of mitochondrial integrity in midbrain DA neurons.** To broadly assess the integrity of the nigrostriatal pathway, ventral midbrain and striatal sections of 16- to 18-month-old mice were analyzed by anti-TH immunohistochemistry. The number of DA neuron cell bodies in the SN and VTA (Fig. 7a, b), as well as the density of TH⁺ fibers in the striatum (Fig. 7c, d), did not show significant

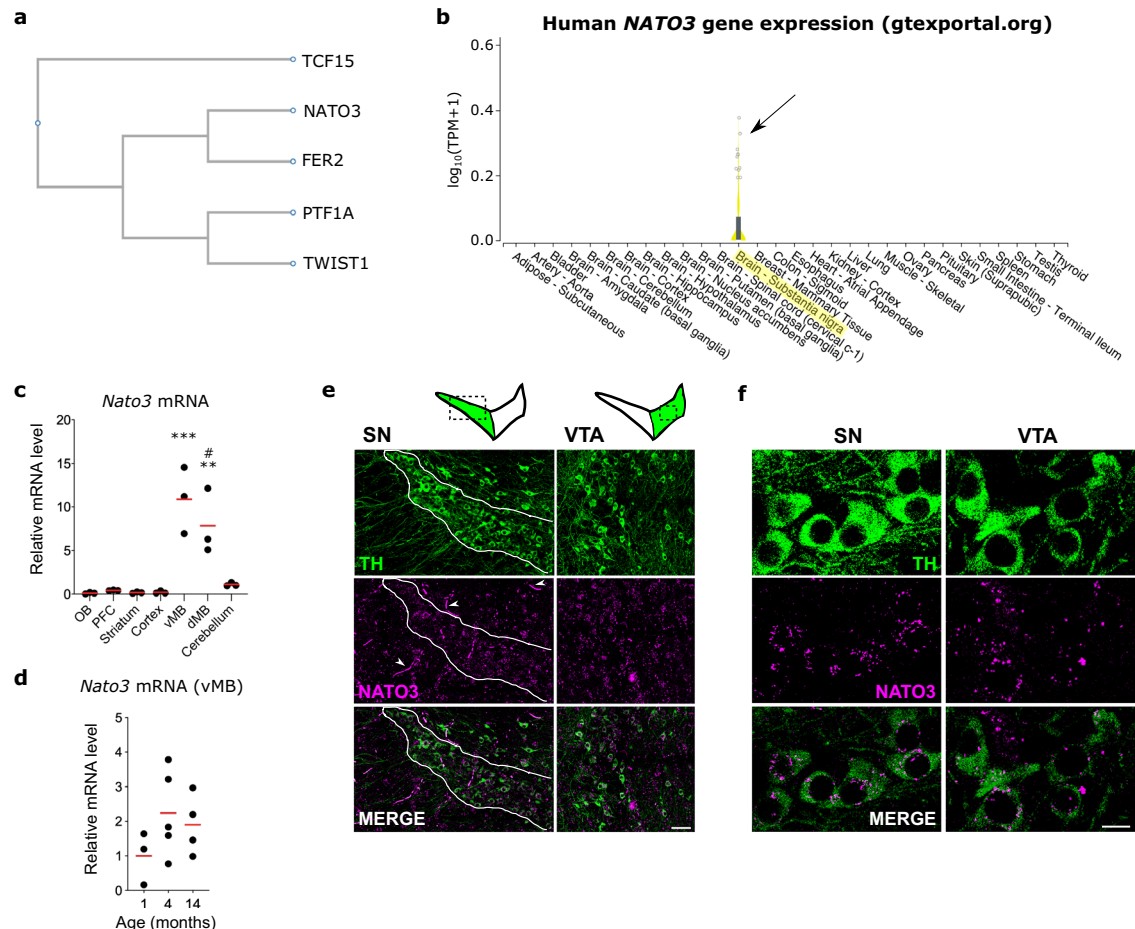

**Fig. 5 *Nato3* is expressed in the midbrain DA neurons in adult mice. a** Phylogenetic tree generated by multiple protein sequence alignment of *Drosophila* FER2 and mouse TCF15, NATO3, PTF1A and TWIST1, using ClustalW (https://www.genome.jp/tools-bin/clustalw). **b** *NATO3* expression levels in human tissues measured by RNA-seq as reported in GTEx portal (gtexportal.org). **c** RT-qPCR analysis of *Nato3* mRNA expression levels in different brain regions of C57BL6 mice at 4 months. OB, olfactory bulbs. PFC, prefrontal cortex. vMB, ventral midbrain. dMB, dorsal midbrain. Data are represented as scatter dot plots, with the horizontal line representing the mean. $n = 3$ animals per region. One-way ANOVA followed by a Tukey's test for multiple group comparison. vMB vs. any region other than dMB, ***$p < 0.001$ (OB vs. vMB, $p = 0.0002$. OB vs. dMB, $p = 0.0050$. PFC vs. vMB, $p = 0.0003$. Striatum vs. vMB, $p = 0.0002$. Cortex vs. vMB, $p = 0.0002$. Cerebellum vs. vMB, $p = 0.0006$). dMB vs. any other region than vMB and cerebellum, **$p < 0.01$ (OB vs. dMB, $p = 0.0066$. PFC vs. dMB, $p = 0.0071$. Striatum vs. dMB, $p = 0.0054$. Cortex vs. dMB, $p = 0.0056$). dMB vs. cerebellum, #$p = 0.0150$. **d** RT-qPCR analysis of *Nato3* mRNA expression levels in the ventral midbrain of C57BL6 mice at different ages. Data are represented as scatter dot plots, with the horizontal line representing the mean of the fold change relative to the expression at 1 month. $n = 3$ (1 month), $n = 5$ (4 months), $n = 4$ (14 months) animals per group. No statistically significant difference between different ages by ANOVA. **e** Representative images of *Nato3* (magenta) in situ hybridization in coronal sections of the ventral midbrain of 4-month-old C57BL6 mice. Arrowheads indicate nonspecific signals. mDA neurons were detected by anti-TH antibodies (green). Scale bar, 50 μm. 3 coronal sections per animal from 2 independent animals were analyzed. **f** High magnification images of *Nato3* (magenta) in situ hybridization of SN and VTA mDA neurons detected by anti-TH antibodies (green). 3 coronal sections per animal from 2 independent animals were analyzed. Scale bar, 10 μm.

differences between control and *Nato3* cKO mice. Analysis of extracellular DA levels in the striatum of 16- to 18-month-old mice by in vivo microdialysis found no difference between control and *Nato3* cKO mice (Fig. 7e). Therefore, locomotor deficits in the *Nato3* cKO mice likely originate from functional impairments of DA neurons, such as changes in electrophysiological properties, rather than an overt loss of cellular integrity.

*Fer2* mutation impairs mitochondrial morphology selectively in the PAM cluster DA neurons, which precedes the degeneration of PAM neurons and locomotor deficits in *Drosophila*[28]. To determine whether *Nato3* ablation also leads to mitochondrial abnormalities in DA neurons in mice, we first visualized mitochondria in the SN using the antibodies against mitochondrial cytochrome c oxidase subunit 4 (COX4). The mitochondria appeared more globular in DA neurons of the SN in 11- and 16-18 months old *Nato3* cKO

mice compared with age-matched controls (Fig. 8a). Quantification of mitochondrial circularity showed a modest but statistically significant increase (i.e., an accumulation of rounded mitochondria) in *Nato3*-ablated DA neurons at both ages. (Fig. 8b). We further examined the ultrastructure and morphology of mitochondria in the SN of 6-month-old and 16-18-month-old mice using transmission electron microscopy. In TH-labeled neurons of control mice at both ages, most mitochondria (>88%) appeared normal, showing a dense and uniform matrix and regularly distributed cristae (Fig. 8c, top panels, and Fig. 8d). Strikingly, whereas most mitochondria were normal in 6-month-old *Nato3* cKO mice, a large fraction (46%) of mitochondria from aged *Nato3* cKO mice displayed an aberrant morphology, showing a reduced and patchy matrix density with few remaining cristae; in certain cases these mitochondria contained almost no cristae, presented white inclusions and appeared

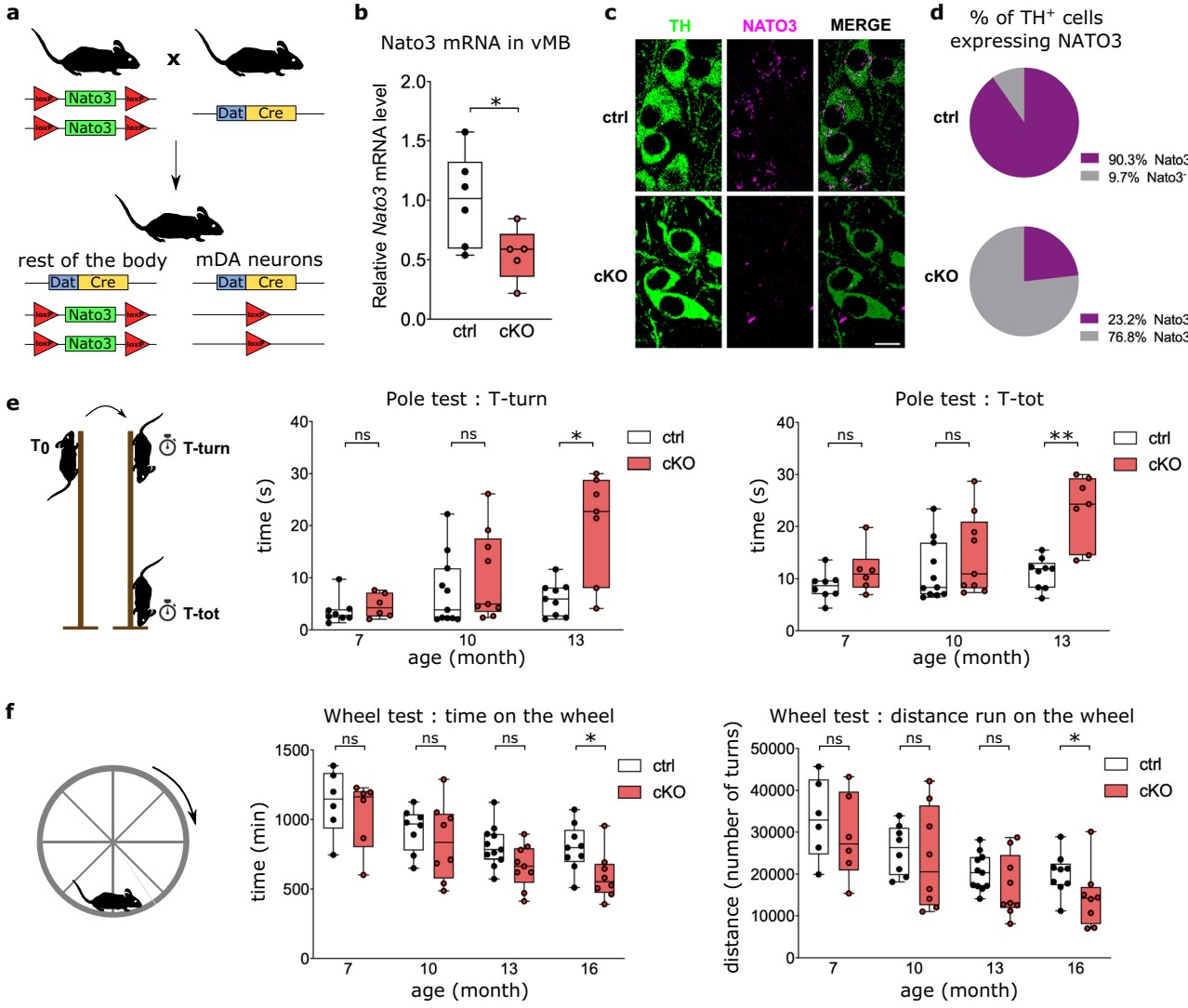

**Fig. 6 Nato3 cKO in DA neurons leads to age-dependent impairment in locomotor coordination and activity. a** Strategy for conditional ablation of *Nato3* in differentiated DA neurons. **b** RT-qPCR analysis of *Nato3* mRNA levels in ventral midbrains of 2-month-old *Nato3floxed* (ctrl) and *Nato3DAT-Cre* (cKO) mice. Data are represented as fold change relative to the average expression level in the ctrl. $n = 5$ (ctrl) and $n = 6$ (cKO) mice. Two-tailed Mann–Whitney test, *$p = 0.0491$. **c** Representative images of *Nato3* (magenta) in situ hybridization of ventral midbrain coronal sections from 4-month-old control and *Nato3* cKO mice. mDA neurons were detected by anti-TH antibodies (green). Scale bar, 10 μm. 3 sections from 2 mice per genotype were analyzed. **d** Quantification of the percentage of TH+ mDA neurons expressing *Nato3* from (**c**). More than 300 TH+ neurons per genotype were analyzed. **e** Pole test. The time taken to orient downward (T-turn; left) and the total time to turn and descend the pole (T-tot; right) are shown. The numbers of animals tested were: $n = 8$ (ctrl, 7 months), $n = 6$ (cKO, 7 months), $n = 11$ (ctrl, 10 months), $n = 9$ (cKO, 10 months), $n = 9$ (ctrl, 10 months), $n = 7$ (cKO, 13 months). Two-tailed Mann–Whitney test comparing ctrl and cKO of the same age. T-turn at 13 months, *$p = 0,0207$. T-tot at 13 months, **$p = 0.0032$. ns not significant. **f** Voluntary wheel running test. The time spent running on the wheel (left) and the total distance run (right) are shown. $n = 6$ animals (ctrl, 7 months), $n = 6$ (cKO, 7 months), $n = 8$ (ctrl, 10 months), $n = 8$ (cKO, 10 months), $n = 11$ (ctrl, 10 months), $n = 9$ (cKO, 13 months), $n = 9$ (ctrl, 16 months), $n = 8$ (cKO, 16 months). Two-tailed Mann–Whitney test comparing ctrl and cKO of the same age. Time at 16 months, *$p = 0.0206$. Distance at 16 months, *$p = 0.0360$. ns not significant. Box boundaries in (**a**), (**e**) and (**f**) are the 25th and 75th percentiles, the horizontal line across the box is the median, and the whiskers indicate the minimum and maximum values.

dysmorphic (Fig. 8c bottom panels and 8d). Mitochondria from TH-negative cells appeared normal in aged *Nato3* cKO mice, suggesting that the age-dependent deterioration of mitochondrial morphology was cell-autonomously induced by *Nato3* deletion in DA neurons (Fig. 8c right bottom panel). Although mitochondrial size did not differ between the two genotypes at both ages, mitochondria in the SN neurons in aged *Nato3* cKO mice were rounder than those in the control mice (Fig. 8e). These results demonstrate that *Nato3* is required for the maintenance of mitochondrial integrity in the DA neurons of the SN during aging. Since mitochondrial dysmorphism appeared at 11 months in *Nato3*

cKO mice (Fig. 8a, b), prior to the onset of locomotor dysfunction (13 months) (Fig. 6e, f), these results also suggest that mitochondrial abnormalities in the SN DA neurons are likely causally involved in the locomotor deficits caused by *Nato3* conditional ablation.

Since *Fer2* loss of function downregulates *Sod2* mRNA levels and SOD2 overexpression partially rescues the loss of PAM neurons in *Fer2²* *Drosophila* mutants (Fig. 4h), we wondered whether SOD2 is downregulated in *Nato3* cKO mice. Anti-SOD2 immunostaining indeed found a significant reduction in SOD2 levels in DA neurons in the VTA and SN of 11- and 16–18 months old *Nato3* cKO mice compared with age-matched controls (Fig. 8f, g). However, *Sod2*

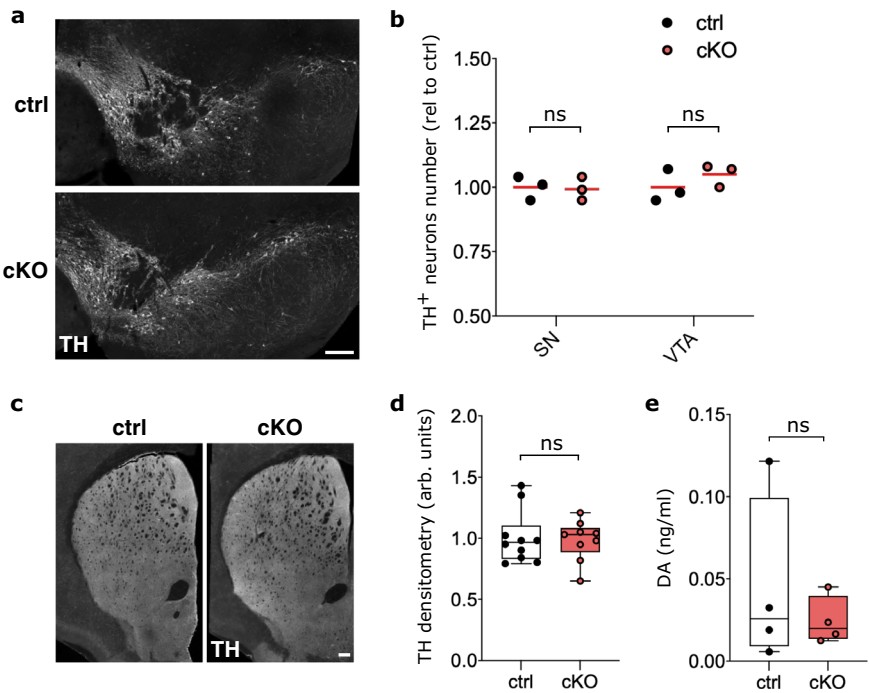

**Fig. 7 *Nato3* cKO in DA neurons does not affect neuronal integrity and striatal DA levels. a** Representative TH immunostaining images of coronal ventral midbrain sections from 16- to 18-month-old control and *Nato3* cKO mice. Scale bar, 200 μm. **b** Quantification of TH⁺ neurons in SN and VTA from (**a**). Data are represented as scatter dot plots, with the horizontal line representing the mean of the fold change relative to the ctrl. $n = 3$ mice per genotype. No statistically significant difference between ctrl and cKO by two-tailed Mann-Whitney test. **c** Representative TH immunostaining images of coronal striatal sections from 16- to 18-month-old control and *Nato3* cKO mice. Scale bar, 200 μm. **d** Optical densitometry from TH immunostaining of the dorsal striatum in (**c**). Data are represented as fold change relative to the expression level in the ctrl. $n = 10$ sections (ctrl) and $n = 9$ sections (cKO) from 4 mice per genotype. No statistically significant difference between ctrl and cKO by two-tailed Mann–Whitney test. **e** Basal levels of extracellular DA (ng/μl) in microdialysis perfusate from striatum of control and *Nato3* cKO mice. $n = 4$ mice per genotype. No statistically significant difference between ctrl and cKO by two-tailed Mann–Whitney test. Box boundaries in (**d**), and (**e**) are the 25th and 75th percentiles, the horizontal line across the box is the median, and the whiskers indicate the minimum and maximum values.

mRNA levels were not different between the two genotypes, indicating that SOD2 levels are post-transcriptionally controlled downstream of *Nato3* (Fig. 8h). These observations suggest that *Fer2* and *Nato3* also share the role of protecting mitochondrial functions through maintaining the antioxidant SOD2 levels, although by different mechanisms.

Several transcription factors involved in mDA neuron development are neuroprotective in postmitotic mDA neurons in the adult brain[16]. Therefore, to explore the molecular mechanism linking *Nato3* ablation and mitochondrial dysfunction in mDA neurons, we analyzed the expression levels of a panel of neuroprotective "developmental" transcription factors in the vMB by RT-qPCR. Interestingly, *Engrailed 1* (*En1*) was significantly downregulated in *Nato3* cKO mice. *Lmx1b* mRNA levels were also reduced in *Nato3* cKO compared to control, but the difference did not narrowly reach statistical significance ($p = 0.0571$) (Fig. 8i). *En1* regulates expression of key proteins of mitochondrial complex I in mDA neurons[20], and *Lmx1b* is required for autophagic-lysosomal pathway[9] and mitochondrial functions[10]. Taken together, our results suggest the possibility that *Nato3* regulates the expression of neuroprotective transcription factors, through which it controls mitochondrial function and morphology in mDA neurons.

## Discussion

Our results demonstrate that *Fer2* is a master regulator of a transcriptional network that maintains mitochondrial integrity in DA neurons in *Drosophila*. This property renders *Fer2* a potent

dopaminergic neuroprotective factor against genetic and oxidative insults implicated in PD pathogenesis. Furthermore, we show that *Nato3*, a mammalian homolog of *Fer2*, is required for the maintenance of mitochondrial integrity in mDA neurons. Both *Fer2* and *Nato3* are developmental transcription factors for DA neurons that continue to be expressed throughout life, and their loss of function in differentiated DA neurons leads to locomotor deficits in aged animals. Our findings corroborate the critical role of developmental transcription factors in the lifelong maintenance of DA neurons and suggest the evolutionary conservation of transcriptional mechanisms for dopaminergic neuroprotection and mitochondrial integrity.

*Drosophila Fer2* (*48 related 2*, CG5952) and its paralogs *Fer1* and *Fer3* were originally named for their similarity to the p48 subunit of the pancreas transcription factor 1 complex (PTF1) in the bHLH domain by the group of Yuh Nung Jan[58]. Independently, due to the similarity to mammalian Atonal within the bHLH domain, *Fer3* (CG6913) was named *Drosophila Nephew of Atonal 3* (*Dnato3*) by the group of Nissim Ben-Arie[59]. Ben-Arie's group also isolated a mouse homolog of *Dnato3* and named it *MNato3*[59]. Brian L Black's lab also independently isolated the same mouse gene and named it *Neuronal-twist* (*N-twist*), due to its high sequence similarly to *M-twist*[60]. It is puzzling that, whereas Jan's group has reported the expression of *Fer2* mRNA in the brain and ventral nerve cord (VNC) of the embryo and *Fer3* expression in the primordial midgut of the embryo[58], Ben-Arie's and Black's groups have shown the expression of *Fer3* mRNA in the CNS of the embryo, which supports the idea that *Fer3* is the homolog of *Nato3*[60].

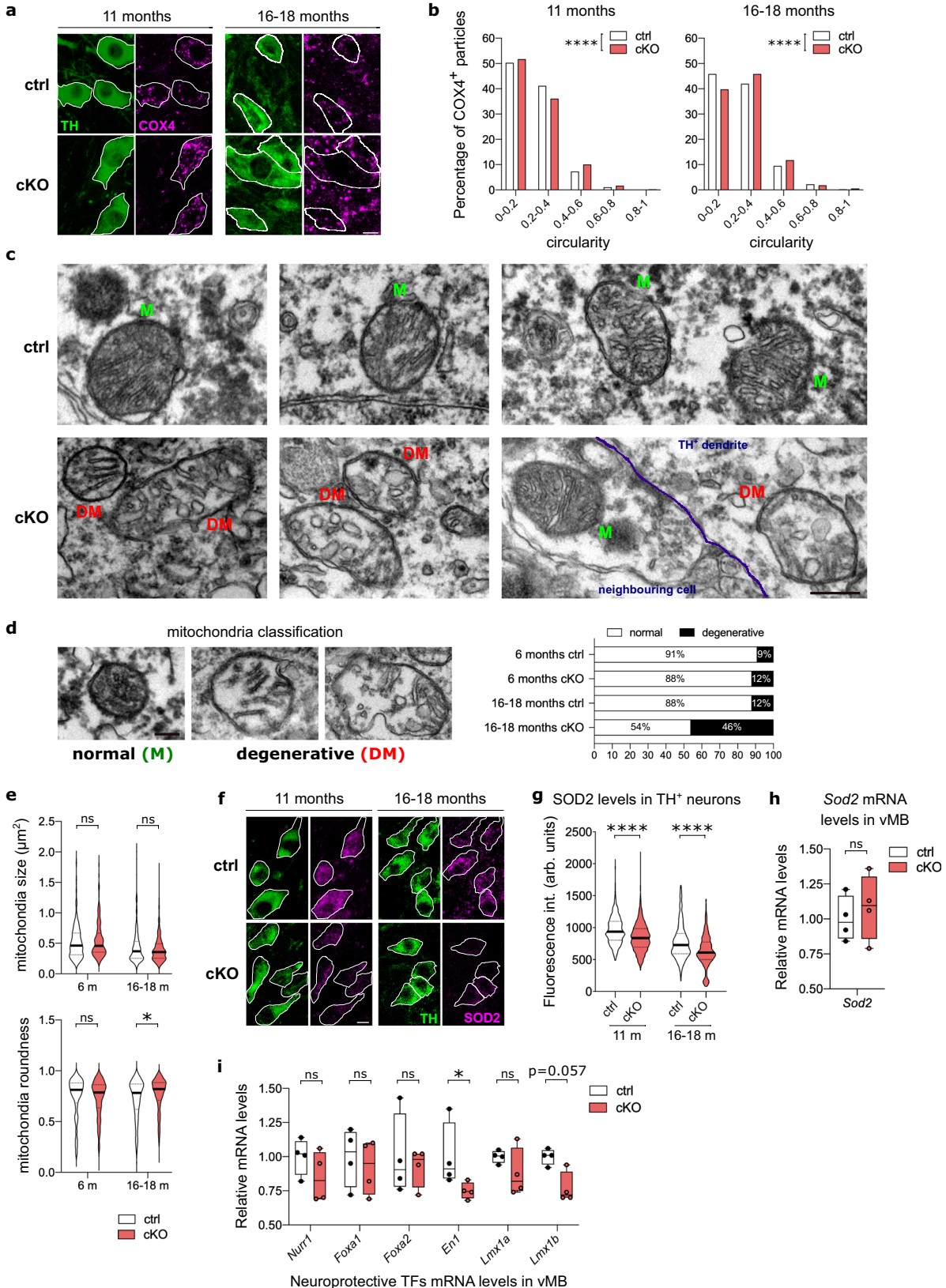

Notwithstanding this inconsistency, we argue that *Fer2* is a functional homolog of *Nato3* based on the following reasons. The fact that fewer DA neurons are generated in *Fer2* loss-of-function mutants[28] is strikingly similar to the observation that 30% fewer mDA neurons are formed in *Nato3* constitutive KO[32], suggesting that *Fer2* bears a proneural function similar to that of *Nato3*.

FER2 bHLH domain shows a high similarity to that of NATO3 as well as to FER1 and FER3 (Supplementary Fig. 4a). Moreover, we have demonstrated here and in previous work[28] that DA neurons are the primary site of *Fer2* expression and function, and that both *Fer2* and *Nato3* are required for mitochondrial maintenance in differentiated DA neurons. Nevertheless, the possibility that

**Fig. 8 *Nato3* is essential for mitochondrial integrity in midbrain DA neurons. a** Mitochondria were visualized by anti-COX4 immunostaining (magenta) in TH[+] neurons (green) of SN in 11- and 16- to 18-month-old control and *Nato3* cKO mice. Scale bar, 10 μm. **b** Distribution of the circularity of COX4[+] mitochondria particles in TH[+] neurons from (**a**). Two-tailed Kolmogorov-Smirnov test, ****$p < 0.0001$. 700 COX4[+] particles per group were analyzed from 3 brains per genotype. **c** Electron micrographs of mitochondria from TH-labeled DA neurons from the SN of 16- to 18-month-old control and *Nato3* cKO mice. M, normal mitochondria. DM, degenerative mitochondria. $n = 3$ mice per group. Scale bar, 0.5 μm. **d** Classification criteria for mitochondria integrity (left) and mitochondria distribution in TH[+] neurons in 6-month-old and 16- to18-month-old control and *Nato3* cKO mice (right). 200 to 350 mitochondria from TH[+] neurons in SN from 3 mice per group were analyzed. Scale bar, 0.25 μm. **e** Measurement of size (top) and roundness (bottom) of mitochondria. 200 to 350 mitochondria from SN TH[+] neurons from 3 mice per group were analyzed. The horizontal line across the violin plot is the median. One-way ANOVA followed by a Tukey's test for multiple group comparison, *$p = 0.0280$. ns, not significant. **f** Representative images of anti-SOD2 (magenta) immunostaining in ventral midbrain coronal sections of 11- and 16 to 18-month-old control and *Nato3* cKO mice. mDA neurons were detected by anti-TH antibodies (green). Scale bar, 10 μm. **g** SOD2 fluorescence intensity in TH[+] neurons from (**f**) were quantified. The horizontal line across the violin plot is the median. 350 (ctrl and cKO at 11 months) and 450 (ctrl and cKO at 16–18 months) TH[+] neurons from 3 brains per group were analyzed. Two-tailed Mann–Whitney test, ****$p < 0.0001$. **h, i** RT-qPCR analysis of *Sod2* (**h**) and neuroprotective transcription factors (**i**) mRNA relative levels in ventral midbrains of 16- to 18-month-old control and *Nato3* cKO mice. $n = 4$ mice per genotype. Box boundaries are the 25th and 75th percentiles, the horizontal line across the box is the median, and the whiskers indicate the minimum and maximum values. Two-tailed Mann–Whitney test, *$p < 0.0286$. ns not significant.

*Fer3* may also have similar roles as *Fer2* in the development and maintenance of DA neurons cannot be excluded and is worthy of further investigation.

This study reports the function of *Nato3* in differentiated mDA neurons in mice, which has not been demonstrated in previous studies. *Nato3* is classically known as a neurogenic transcription factor, expressed in the floor plate cells in the developing mesencephalon and spinal cord[32,34]. Although *Nato3* expression in the spinal cord of postnatal and adult mice has been reported[61], we found here its presence in the mDA neurons of adult mice. By conditionally ablating *Nato3* in postmitotic DA neurons, we were able to distinguish the function of *Nato3* in mature mDA neurons, i.e., the maintenance of mitochondrial integrity, from its role in neurogenesis. Importantly, human *Nato3* is expressed almost exclusively in the SN among adult tissues (Fig. 5b) (the GTEx Consortium, Nat Genetics, 2013)[53]. Thus, it is plausible that human *Nato3* also plays an important role in the mitochondrial maintenance in SN dopaminergic neurons.

The genesis, specification and differentiation of mDA neurons are governed by a complex web of transcription factors[62]. Whereas *Nato3* acts as an "early transcription factor" required for neurogenesis[32], it can also directly regulate *Lmx1b*, *Shh* and *Foxa2* and thereby indirectly regulates the "late transcription factor" *Nurr1*[33,52]. Since *Lmx1b*, *Foxa2* and *Nurr1* are neuroprotective transcription factors[9,12,13,19], *Nato3* might contribute to dopaminergic neuroprotection by regulating some of these genes. Indeed, we find here that *En1* and *Lmx1b* are downregulated in the vMB of *Nato3* cKO mice, supporting this possibility. Furthermore, a recent study has shown that *Nato3* is a potential target of the transcription factor sterol regulatory element binding protein 1 (SREBP1), a key player of mDA neurogenesis[63]. SREBP1 also upregulates *Foxa2*[63], which in turn positively regulates *Nato3* expression during development[64]. Importantly, a polymorphism in the *Srebf1* gene encoding SREBP1 is linked to PD[65,66]. Therefore, the evidence is growing that gene regulatory programs involving DA developmental genes, including *Nato3*, have important roles in the survival of mDA neurons during aging and offer an opportunity to study PD.

Identification of *Nato3* transcriptional targets and downstream molecular events in differentiated mDA neurons will provide crucial information to understand the role of *Nato3* in mitochondrial maintenance. While further studies are required to obtain such data, here we have performed homologous experiments in *Drosophila*, i.e., identification of direct and indirect targets of *Fer2* in DA neurons in the PAM cluster. An intersectional approach combining ChIP-seq, whole-head RNA-seq and RNA-seq of isolated PAM neurons, unequivocally identified 26 direct and over 100 indirect *Fer2* target genes expressed in PAM neurons. Among *Fer2* indirect targets, genes involved in mitochondrial function and structure, including those required for ETC complexes assembly, are remarkably enriched. Malfunction of the ETC leads to ROS overproduction, which in turn induces mitochondrial DNA mutations, alters membrane permeability and mitochondria morphology, and further damages the ETC complexes, in a pathogenic self-reinforcing loop. Indeed, dysfunctions of the ETC, particularly of Complex I, have been proposed as a leading mechanism of DA neurodegeneration[67,68]. Thus, our molecular data are in accordance with the observation that *Fer2*[2] flies display an overt defect in mitochondrial networks[28].

It is noteworthy that *Sod2*, which encodes mitochondrial superoxide dismutase, is also an indirect target of *Fer2*. Consistent with this finding, *Sod2* overexpression partially rescues the loss of PAM neurons in *Fer2*[2] mutants. Furthermore, *Fer2* overexpression reverses mitochondrial abnormalities in the PAM neurons of *park*[1] mutants and prevents PAM neuron loss in genetic and toxin models of PD. These results are in agreement with previous reports that *Sod2* and *Sod2*-mimetic agents are protective in cell and fly models of PD[69], and endorse the further exploration of *Sod2* as a potential therapeutic target for PD. Interestingly, SOD2 levels in the mDA neurons are controlled by *Nato3*, although not by a transcriptional mechanism. These findings highlight the conservation of the neuroprotective role played by *Fer2* in flies and *Nato3* in mice.

Among *Fer2* direct targets, we found six genes whose dysregulation induces a loss of PAM neurons. None of these genes have been previously related to the DA system or implicated in neuronal development or survival. Further studies are needed to clarify how these actors and their mammalian homologues contribute to the biology of DA neurons. The number of genes implicated in PAM neuron survival may increase in the future, as our list of *Fer2* direct targets is likely not exhaustive. *Fer2* is a stress-responsive transcription factor that is upregulated by oxidative insult or possibly by other stressors[28]. Under such conditions FER2 may be recruited to and/or regulate different sets of target genes.

A large fraction of *Fer2* primary targets encode transcription factors or chromatin proteins. This positions *Fer2* at the vertex of multiple transcriptional cascades and suggests that *Fer2* orchestrates the expression of a broad set of secondary targets (Supplementary Fig. 3b). *Fer2* expression is restricted to PAM neurons and a few other cell clusters in the brain, and ubiquitous or even

pan-neuronal constitutive overexpression of *Fer2* causes lethality (data not shown), indicating that genetic programs driven by *Fer2* are highly specific to PAM neurons.

PD progresses from the asymptomatic (preclinical) period to a stage with a variety of nonmotor symptoms and subtle motor impairments that do not meet diagnostic criteria (prodromal PD), before a definite diagnosis is reached[70]. Modeling in animals this latent phase of the disease would facilitate the development of disease-modifying therapies and the discovery of early PD biomarkers. *Nato3* cKO mice reproduce some features of prodromal PD, presenting moderate motor deficits and quasi-normal mDA neurons that accumulate abnormal mitochondria. Striatal extracellular DA concentration in *Nato3* cKO mice are not significantly different from that of control mice at 16–18 months, when *Nato3* cKO mice manifest locomotor deficits. Mitochondrial dysmorphism and, moreover, downregulation of SOD2 precedes the onset of locomotor deficits in *Nato3* cKO mice, suggesting that mitochondrial impairment may be causally linked to locomotor dysfunction. In line with our findings, a recent study has demonstrated that disruption of mitochondrial complex I in differentiated DA neurons is sufficient to cause progressive motor dysfunction[71]. Therefore, functional impairments in mDA neurons following mitochondrial disfunction, such as metabolic reprograming from mitochondrial oxidative phosphorylation (OXPHOS) to glycolysis, axonal dysfunction[71], changes in electrophysiological properties[72] or in monoamine metabolism[73], might account for motor deficits in *Nato3* cKO mice. Likewise, other PD mouse models, such as *PINK1* deficiency and *Parkin* deletion mice, do not show loss of DA neurons or reduction in nigrostriatal DA release, while exhibiting mitochondrial impairments and locomotor phenotypes[37,73]. There is growing evidence that neurodegeneration in PD is manifested after several risk factors, including genetic predisposition, toxin exposure and aging, interact to overcome protective compensatory mechanisms[74]. Future work is needed to determine whether exposing *Nato3* cKO mice to a second genetic or environmental insult exacerbates its PD-related phenotypes and trigger overt DA neurodegeneration.

The activation of neuroprotective transcription factors offers potent therapeutic opportunities, allowing to modulate multiple downstream pathways in parallel. Indeed, *Nurr1* and *Foxa2* gene delivery[19], *Nurr1* activators[75] and *En1* infusion[20] in rodent PD models have shown encouraging results. In light of our findings that *Fer2* confers neuroprotection in multiple fly PD models, it is tempting to speculate that the overexpression or activation of *Nato3* may protect mDA neurons from genetic and environmental insults in mammals, with potential far-reaching implications for PD treatment.

## Methods

**Drosophila stocks and maintenance**. The following lines were obtained from the Bloomington *Drosophila* Stock Center (BDSC) (Indiana University, Bloomington, IN): *UAS-RedStinger* (#8546), *UAS-Sod2* (#24494), *park¹* (#34747), and *hs-GAL4* (#1799). *Fer2::GFP::V5, PBac{RB}Fer2^{e03248}* (referred to as *Fer2¹*), *Mi{ET1} Fer2^{MB09480}* (referred to as *Fer2²*), *Fer2-GAL4, R58E02-GAL4, UAS-mitoGFP*, and *UAS-Fer2::FLAG* lines were previously described[26,28,76]. *UAS-Lrrk WT* and *UAS-Lrrk I1915T* were a kind gift from Yuzuru Imai[77]. All of the lines used to over-express or knock-down *Fer2* primary targets are described in the Supplementary Data 3. All flies were maintained on standard cornmeal-agar food at 25 °C in a 12 h:12 h light-dark cycle and under controlled humidity.

**Fly brain immunohistochemistry**. Immunostaining of whole male fly brains was performed as previously described[28]. Briefly, the flies were decapitated, and the heads were fixed in 4% paraformaldehyde (PFA) + 0.3% Triton X-100 for 1 h on ice and washed three times with PBST-0.3 (PBS, 0.3% Triton X-100). The cuticle was partially removed, and the partly opened heads were incubated with blocking solution (5% normal goat serum, PBS, 0.3% Triton X-100) for 1 h at room temperature (RT) on a rocking platform. Subsequently, the samples were incubated with primary antibodies for 48 h at 4 °C. After three 20 min washes with PBST-0.3, the samples were then incubated with Alexa Fluor-conjugated secondary antibodies overnight at 4 °C. After three 20 min washes with PBST-0.3, the remaining cuticle and trachea were removed, and the brains were mounted in Vectashield medium (Vector Laboratories). Primary antibodies and the dilution factors were as follows: rabbit polyclonal anti-TH (ab152, Millipore) 1:100, polyclonal rabbit anti-GFP (A6455, Invitrogen) 1:250. Neuropil was counterstained with the mouse monoclonal antibody nc82 (Developmental Studies Hybridoma Bank) 1:100 to visualize the entire brain and assess its integrity. Secondary antibodies and the dilution factors were as follows: goat anti-rabbit IgG Alexa Fluor 488 conjugate (A-11034, Thermo Fisher Scientific) 1:250, goat anti-mouse IgG Alexa Fluor 633 conjugate (A-21052, Thermo Fisher Scientific) 1:250.

**Mouse brain immunohistochemistry**. Male mice were anesthetized and perfused through the left ventricle with 0.2% Heparin in PBS at RT followed by ice-cold 10% formalin solution (Sigma HT501128-4L). The brains were dissected and post-fixed for 72 h in fresh 4% PFA (Thermo Fisher Scientific #28906) before being embedded in low melting point agarose (Thermo Fisher Scientific #R0801) and sectioned in a sliding vibratome at a thickness of 40 μm. Striatum sections were stained as floating sections. Ventral midbrain sections were adhered to the SuperFrost plus slides (Thermo Fisher Scientific J1800AMNZ) before staining. Slices were incubated in blocking buffer (4% non-fat milk, 4% Bovine Serum Albumin (BSA), PBS) containing 0.1% Triton X-100 for 30 min at RT. The sections were then incubated with primary antibodies in blocking buffer containing 0.1% Tween for 48 h at 4 °C. After three 30 min washes in washing buffer (0.1% Tween, PBS), sections were incubated with the corresponding secondary antibody overnight at 4 °C, in the dark. After three 30 min washes in washing buffer, the slices were mounted with DABCO-glycerol mounting medium (25 mg/ml DABCO (Sigma D27802-25G), 90% glycerol, PBS; pH 8.6). For nuclear staining, prior to mounting midbrain slices were treated with 100 μg/ml RNAse A (#EN0531, Thermo Fisher Scientific) for 30 min at 37 °C, and then incubated with TO-PRO-3 iodide (T3605, Sigma) for 30 min at RT. Primary antibodies and dilution factors were as follows: rabbit polyclonal anti-TH (Millipore ab152) 1:300, mouse anti-TH (Immunostar 22941) 1:300, mouse anti-COX4 (Thermo Fisher Scientific 1D6E1A8) 1:300, rabbit anti-SOD2 (abcam ab13534). Secondary antibodies and dilution factors were the same as those for the fly brain immunostaining.

**Microscopy and image analysis**. Fly brains and mouse brain slices were scanned using a Leica TCS SP5 confocal microscope. Image analysis was performed using Fiji software[78]. All analyses were conducted blindly with respect to genotype.

*Fly brain*. TH-positive neurons were counted manually from Z-stacks of confocal images using the cell counter plugin of Fiji. PAM neuron counts were scored blindly. *Fer2::GFP::V5* and TH fluorescence intensity in the PAM region was measured on Z-SUM projection images. The region of interest (ROI) corresponding to the PAM cluster was manually drawn to include TH⁺ neurons. Background signal measured outside the ROI was subtracted from the ROI signal. To quantify mitochondria circularity in PAM clusters, mitoGFP-positive objects within the PAM neurons region were analyzed using a particle analysis plugin as described previously[29].

*Mouse brain*. TH immunoreactivity in striatal regions was quantified by optical densitometry using Fiji. Three striatum images were acquired for each animal, and in every picture fluorescence intensity was measured for four squares drawn on the dorsal striatum ($n = 3$–$4$ animals per genotype). Measures from all the squares were averaged to give the value for each animal. To obtain the number of mDA neurons, TH-positive neurons were counted manually from Z-stacks of confocal images using the cell counter plugin ($n = 3$ animals per genotype). Every fourth section covering the entire VTA and SN in one brain hemisphere was included in the counting procedure. Neuron counts were scored blindly. To quantify mitochondria circularity in DA neurons, COX4-positive objects within the TH-positive DA neurons were analysed using the particle analysis plugin.

**Treatment of flies with oxidative reagents**. 7-day-old male flies were placed into empty vials and starved for 6 h. The flies were then transferred into vials containing *Drosophila* instant food (Formula 4-24 Instant Drosophila Medium, Carolina Biological Supply Company, USA) containing 5% $H_2O_2$ (Sigma, H1009), or vials containing *Drosophila* instant food only as a control, and kept for 24 h. All vials were kept in a humid box at 25 °C during the treatment. Following the treatment, flies were transferred back to the vials with standard cornmeal-agar food and maintained normally. Seven days after the treatment the flies were decapitated, and anti-TH-staining was performed as described above.

**ROS detection**. The level of ROS production in the male fly brain was measured using a cell-permeable probe, 2'7'-dichlorofluorescin diacetate (H2DCF), following the protocol described in[79]. The fluorescence intensity for the entire central brain was measured using Fiji software[78] after generating a Z-SUM projection across the corresponding Z-stacks, as previously described[28].

**TMRM staining**. 14-day-old male flies expressing *UAS-GFP* driven with *R58E02-GAL4* were dissected in PBS. Fly brains were incubated with 100 nM Tetramethylrhodamine methyl ester (TMRM) (Invitrogen, I34361) for 30 min at 37 °C and washed three times with Hank's balanced salt solution (HBSS). Brains were mounted on slides with warm HBSS. TMRM was excited with 561 nm and the fluorescence emission above 580 nm was measured. To quantify TMRM signal within PAM neurons, PAM neurons were defined by thresholding the GFP signal using Fiji software[78]. TMRM fluorescence intensity was measured within the defined volume in a SUM projection of respective z-stacks.

**Startle-induced climbing assay**. For each experimental group, 20 male flies were briefly anesthetized with $CO_2$ and placed into a 100 ml graduated glass cylinder, which was divided into five equal parts marked 1–5 from bottom to top. Flies were left to recover from the $CO_2$ exposure for 1 h before the assay. The climbing assay was performed at ZT2 (2 h after lights-on in 12 h:12 h light-dark cycles) to avoid circadian variations in the locomotor activities of the flies, following the procedure previously described[28]. Briefly, the cylinder was tapped on a flexible pad to collect all of the flies at the bottom, and then the flies were allowed to climb up the wall of the cylinder for 20 s. The number of flies that climbed up to each zone within 20 s was counted manually and used to calculate the climbing index (CI) with the following formula: $CI = (0 \times n_0 + 1 \times n_1 + 2 \times n_2 + 3 \times n_3 + 4 \times n_4 + 5 \times n_5) / n_{total}$, where $n_x$ is the number of flies that reached zone X, $n_0$ is the number of flies that remained in the bottom of the cylinder, and $n_{total}$ is the total number of flies. Each experiment consisted of three trials, and three independent experiments were performed for each genotype. The mean CI values from three independent experiments are shown.

**Chromatin immunoprecipitation coupled to sequencing (ChIP-seq)**. ChIP was performed based on previous protocols[80,81] on male and female *Fer2[1]* mutant flies expressing a GFP::V5-tagged *Fer2* genomic transgene (*Fer2::GFP::V5*) and on *w[1118]* flies (negative control). Briefly, ~8000 14-day-old flies per sample were collected 2 h before lights-on (ZT22), frozen in liquid nitrogen and maintained at −80 °C until processed. Heads were collected using metal sieves, and then 1 ml of heads was ground in liquid nitrogen and homogenized 30 times in 5 ml of NE Buffer (15 mM HEPES pH 8, 10 mM KCl, 0.1 mM EDTA, 0.5 mM EGTA, 350 mM sucrose, 0.1% Tween 20, 5 mM MgCl2, 1 mM DTT, 1 mM PMSF plus Protease inhibitor cocktail (Roche 11 873 580 001)) with 1% PFA. Fixation was performed for 10 min at RT and quenched with 125 mM glycine for 5 min at RT. The homogenate was filtered through a 100 μm nylon mesh filter (Thermo Fisher Scientific 22363549), and nuclei were collected by centrifugation at $800 \times g$ for 5 min at 4 °C. The nuclei were washed three times, resuspended in 1.5 ml of cold RIPA buffer (25 mM Tris-HCl pH 8, 150 mM NaCl, 0.5% sodium deoxycholate, 0.1% SDS, 1% NP-40, 0.5 mM DTT plus Protease inhibitor cocktail) and sonicated using a Bioruptor sonicator for 18 min (30 s ON/30 s OFF). The sonicated chromatin was then centrifuged at $10,000 \times g$ for 10 min at 4 °C to remove cell debris. The supernatant was pre-cleared with agarose beads (1 h, 4 °C) and incubated with 50 μl of saturated anti-V5 beads (Sigma A7345) overnight at 4 °C. The beads were washed twice with low-salt buffer (20 mM Tris pH 8, 150 mM NaCl, 2 mM EDTA, 0,1% SDS, 1% Triton X-100), twice with high-salt buffer (20 mM Tris pH 8, 500 mM NaCl, 2 mM EDTA, 0.1% SDS, 1% Triton X-100), twice with LiCl buffer (10 mM Tris pH 8, 250 mM LiCl, 1 mM EDTA, 1% sodium deoxycholate, 1% NP-40) and twice with TE buffer (10 mM Tris-HCl pH 8, 1 mM EDTA). DNA was eluted in the V5 elution buffer (10 mM HEPES, 1.5 mM MgCl2, 0.25 mM EDTA, 20% glycerol, 250 mM KCl, 0,3% NP-40, 0.5 mg/ml V5 peptide) and incubated with RNase (30 min, 37 °C) and proteinase K (overnight, 65 °C). Subsequently, the DNA was purified with a MinElute PCR purification kit (Qiagen 28004). Libraries were prepared with 0.5 to 0.7 ng of ChIP-enriched DNA as starting material and processed using the Illumina TruSeq ChIP kit according to the manufacturer's specifications, at the iGE3 Genomics Platform in Geneva. The libraries were validated on a Tapestation 2200 (Agilent) and a Qubit fluorometer (Invitrogen - Thermo Fisher Scientific). Single reads of 50 bp were generated using the TruSeq SBS HS v3 chemistry on an Illumina HiSeq 2500 sequencer.

**ChIP-seq analysis**. Sequencing reads were mapped to the *Drosophila melanogaster* genome reference dm3 using Bowtie2 with default settings[82]. *Fer2* binding sites were identified by individually comparing each FER2 ChIP sample ($N = 2$) to each $w^{1118}$ ChIP sample ($N = 2$; negative control) using MACS2[83]. The minimum FDR (q value) cutoff for peak detection was 0.05. The 267 shared peaks overlapping in all the four FER2 ChIP vs. $w^{1118}$ ChIP comparisons were selected as confident peaks. Peak calling was also performed using Homer[84] and over 90% of Homer peaks overlapped with MACS2 peaks, confirming our results. ChIP-seq profiles were visualized using IGV[85]. HOMER motif analysis algorithm was used to search for enriched motifs, and identified a 12-mer containing a bHLH binding site, present in the majority of the best ranked peaks. Peak-to-gene assignment was performed using the HTSstation[86] and resulted in a list of 279 genes.

**Whole-head RNA-seq**. *hs-GAL4, UAS-Fer2::FLAG* male and female flies (*hs > Fer2::FLAG*) were heat-shocked at 37 °C for 3 h, or maintained at 25 °C (control), and collected 12 h or 48 h later (three independent biological replicates per group).

The same procedure was also applied to flies carrying *hs-GAL4* but not the *UAS-Fer2*, in order to identify and exclude from our analysis the genes that responded to heat irrespective of *Fer2* induction. The flies were frozen in liquid nitrogen and kept at −80 °C until processed. Fly heads were collected using metal sieves and kept in Trizol (Life Technologies 15596026) at −80 °C until processed. Total RNA was isolated from dissected mouse brain regions using Trizol according to the manufacturer's instructions and treated with Turbo DNase for 30 min at 37 °C (Turbo DNA-free kit, Thermo Fisher Scientific AM1907). Libraries were prepared at the iGE3 Genomics Platform in Geneva. Single reads of 50 bp were generated using an Illumina HiSeq 2500 sequencer. Data analysis was performed using Galaxy[87]. Sequencing reads were mapped to the *Drosophila melanogaster* genome reference dm3 using TopHat[88] with default settings. Differentially expressed genes (DEG) analysis was performed with Cufflinks, Cuffmerge and Cuffdiff, to compare flies who received the heat shock treatment with control flies kept at 25 °C. The same analysis was performed on *hs-GAL4* flies, to identify the genes which responded to heat irrespective of *Fer2* induction. These heat-responsive genes were subtracted from DEG identified in the *hs > Fer2::FLAG* experiment.

**PAM-specific RNA-seq**. Flies carrying *R58-GAL4* were combined with *UAS-Fer2::FLAG* to overexpress *Fer2*, and with *UAS-RedStinger* to mark PAM neurons. Flies carrying only *R58-GAL4* and *UAS-RedStinger* were used as a control. Cells were dissociated from the brains of ~70 14-day-old male and female flies 2 h before lights-on (ZT22) as previously described[76], except that the brains were treated with papain for 25 min. After trituration in 300 μl SMactive medium containing 50 μm AP5, 20 μm DNQX and 0.1 μm TTX, the cell suspension was filtered through a sterile 70 μm membrane and distributed to three wells of three two-well chamber glass slides (LabTek II, Nunc) kept on ice. The second well of each slide contained ice to cool the cells while RedStinger-positive cells were sorted on a fluorescence microscope equipped with a motorized stage and a needle holder (CellSorter Company for Innovations). Cells were selected manually using capillaries with 50 μm tip diameter. Collected cells were released in a clean part of the chamber and immediately picked up again, greatly reducing the amount of debris, before being transferred to a fresh slide with 100 μl medium spread in a 1 × 1 cm GeneFrame (Thermo Fisher Scientific). Cells were reselected for two rounds and finally transferred using a new capillary with 30-μm tip diameter to 100-μl lysis buffer (Absolutely RNA Nanoprep Kit; Agilent). Cells of multiple collections were pooled if needed to obtain 200 cells per sample, and total RNA was extracted according to the manufacturer's instructions. The RNA was converted to cDNA and amplified using the SMART-Seq4 Kit (Takara Bio). Libraries were generated using NexteraXT (Illumina) and sequenced on an Illumina HiSeq 2500 or 4000. DEG analysis was performed using the HTSstation[86] and resulted in a list of 132 genes.

**Bioinformatic analysis of genes found in ChIP-seq and RNA-seq**. Since human databases are generally more comprehensive, fly genes were converted into human orthologs prior to Gene Ontology (GO) analysis[89]. Human orthologs of fly genes were predicted using DIOPT[90] and were analyzed for GO with Metascape[89]. A network representation of GO terms was obtained using Cytoscape[91]. A search for TF binding sites enriched among promoters of PAM-specific RNAseq DEG was performed using PSCAN[92]. Protein-protein interactions were investigated using STRING[93] and a previously reported *Drosophila* TFs database[49].

**Mice strains**. *Nato3-LoxP* mice was generated using Clustered Regularly Interspaced Short Palindromic Repeats (CRISPR) technology by Applied StemCell, Inc. (Milpitas, CA). Two copies of a synthetic *LoxP* sequence were inserted upstream and downstream of the unique exon of *Nato3* gene. Briefly, a mixture of active guide RNA molecules (gRNAs), two single stranded oligo donor nucleotides (ssODNs) and *Cas-9* mRNA was prepared and injected into the cytoplasm of C57BL/6 embryos. Mice born from the microinjection were screened to identify *LoxP* insertions at the designated locations using PCR and sequencing. The founder mice were bred to evaluate the germline transmission and expand the *Nato3-LoxP* colonies. *DAT-Cre* mice have been described previously[54] (obtained from Jean-Claude Martinou's laboratory, University of Geneva). *DAT-Cre* and *Nato3-LoxP* mice were maintained in a C57BL/6 J genetic background. Crosses between these lines generated mice that are homozygous for the *Nato3-LoxP* allele and harbor one copy of the *DAT-Cre* transgene (*Nato3[DAT-Cre]*). Littermates homozygous for *Nato3* floxed alleles and harboring no copy of the *DAT-Cre* transgene were used as controls (*Nato3[floxed]*). Mice were maintained in rooms with controlled 12 h light/dark cycles, temperature between 23–24 °C, and humidity of 47–61%, with food and water provided *ad libitum*. Mice were housed at a maximum of five animals per cage in individually ventilated cages. All experiments were conducted in accordance with the Institutional Animal Care and Use Committee of the University of Geneva and with permission of the cantonal authorities (Permit No. GE/3/19 27381, GE/78/19 31317 and GE/29/33210).

**Electron microscopy image acquisition and analysis**. Male animals were anesthetized, perfused and embedded in agarose as described in the mouse brain immunohistochemistry section, except that the perfusion was performed with 3% PFA complemented with 0.2% glutaraldehyde in 0.1 M phosphate buffer (PB), pH 7.4. Subsequently, 60 μm-thick sections were cut and processed for pre-embedding

immunoperoxidase staining, as previously described[94]. Briefly, immunostaining for DA neurons in the SN was performed by incubating the sections with rabbit anti-TH antibodies (Millipore ab152, 1:100) followed by goat anti-rabbit biotinylated secondary antibodies (Vector Laboratories PK-4001, 1:200) and then avidin biotin peroxidase complex (Vector Laboratories PK-4001). 3,3′-diaminobenzidine tetra-chloride (DAB; Invitrogen Life Technologies 750118) was applied for 1–2 min until the stain developed. Sections were further embedded in EPON resin (Fluka) and trimmed around the labeled cells of interest. Ultrathin (60 nm) sections were produced and imaged by a Tecnai G212 electron microscope (FEI Company, Oregon, United states) equipped with a digital camera (Mega View III; Soft Imaging Systems). ~15 labeled cells from two mice were analyzed for each genotype. Mitochondria were classified blindly. Mitochondrial size and roundness were measured via Fiji software using the particle analysis plugin after a manual selection of each mitochondrion.

**RT-qPCR on mouse brain regions**. Male and female animals were sacrificed by cervical dislocation and decapitated. The brains were immediately removed and all the subsequent steps were performed on ice. The olfactory bulbs, prefrontal cortex, striatum, cortex, and cerebellum were scalpel-dissected. The ventral and dorsal midbrain were scalpel-dissected from 1 mm-thick brain slices cut in an ice-cold coronal matrix (Zivic Instruments). Samples were kept in Trizol (Life Technologies 15596026) at −80 °C until processed. Total RNA was isolated from dissected mouse brain regions using Trizol according to the manufacturer's instructions and treated with Turbo DNase for 30 min at 37 °C (Turbo DNA-free kit, Thermo Fisher Scientific AM1907). Reverse transcription was performed from 500 to 1000 ng RNA using the Maxima reverse transcriptase (Thermo Fisher Scientific 18090050) and oligo-dT primers, according to the manufacturer's instructions. Primers were designed with Primer-BLAST online tool (https://www.ncbi.nlm.nih.gov/tools/primer-blast/): *B2m* (F: 5′-AGAATGGGAAGCCGAACATA, R: 5′-CGTTCTTCA GCATTTGGATT), *Cyclophilin* (F: 5′-TGCCATCCAGCCAGGAGGTC, R: 5′-CCA TCGTGTCATCAAGGACTT), *Hmbs* (F: 5′-GCTGAAAGGGCTTTTCTGAG, R: 5′-TGCCCATCTTTCATCACTGT), *Hprt1* (F: 5′-TGTCAGTTGCTGCGTCCCCA GA, R: 5′-TCTACCAGAGGGTAGGCTGGCC), *Pgk1* (F: 5′-ACCTGCTGGCTGG ATGGGCT, R: 5′-CTCGACCCACAGCCTCGGCA), *Rpl13a* (F: 5′-CGGATGGT GGTCCCTGCTG, R: 5′-GAGTGGCTGTCACTGCCTGG), *Nato3* (F: 5′-GGTGA GCCCAAGAGACACTC, R: 5′-GTGCCGGATCCCCAACTTAT), *Engrailed 1* (F: 5′-ACGCACCAGGAAGCTAAAGA, R: 5′- CTGGAACTCCGCCTTGAGTC), *Foxa1* (F: 5′-GAGAGCCATCATGGTCATGTCA, R: 5′-TGGGGGTGGGGGAAT CCTTTA), *Foxa2* (F: 5′-CCTCAAGGCCTACGAACAGG, R: 5′-GCTTTGTTCGT GACTGGGC), *Lmx1a* (F: 5′-TATACAACGTTGCCCACCCC, R: 5′- GTGATCTC CAGGCATCTGGG), *Lmx1b* (F: 5′-GCAGTGGAGATGACGGGAAA, R: 5′-GAA AGCTCTTCGCTGCTGTG), *Nurr1* (F: 5′-TTCCAGGCAAACCCTGACTAT, R: 5′-TGCCCACCCTCTGATGATCT). Quantitative PCR (qPCR) was performed on the Light Cycler LC480 (Roche) using the TB Green Ex Taq II polymerase (Takara RR820L). Each reaction was performed in triplicate. Relative quantification and normalization were performed with a multiple reference genes model[95]. For normalization, the best reference genes for each experiment were determined using the geNorm algorithm (https://genorm.cmgg.be) within a panel of 4–5 putative reference genes. To compare *Nato3* levels between different brain regions, *B2m* and *cyclophilin* were used as reference genes. To compare *Nato3* levels in the midbrain between different ages, *B2m* and *Hprt1* were used.

**Behavioral tests**. Male mice of different ages (7, 10, 13 and 16 months) were evaluated for posture control and coordination (pole test), spontaneous locomotor activity (wheel test) and exploration-driven locomotor behavior (open field test). All behavioral tests were performed during the light cycle by experimenters blind to the genotype. The mice were allowed to habituate to the behavioral room for at least 45 min before each test. Behavioral equipment was cleaned with 70% ethanol after each test session to avoid olfactory cues.

*Pole test*. The pole test was performed as previously described[56] with minor modifications. Mice were first trained 4–6 times by placing the animal head-down on top of a vertical pole (diameter: 1 cm, height: 55 cm) and letting it descend. Then, animals were trained 3–4 times in the regular turning and descending procedure, by placing the animal head-up on top of the pole. For the actual test, mice performed eight trials and were recorded with a video camera. The time to orient downward (T-turn) and the total time to turn and descend the pole (T-tot) were measured, with a maximum duration of 30 sec. When the mouse was not able to turn downward and instead dropped from the pole in a lateral body position, a value of 30 sec was assigned. The average of the eight trials was used as the final score. During the training and for the real test, mice were tested a maximum of three times per day, with an interval of at least 5 min between trials.

*Running wheel test*. Mice were singly housed in cages with free access to a running wheel (14 cm in diameter) for 6 days, under a 12 h light/dark cycles. After three days of habituation, the number of wheel revolutions per minute was recorded with a sensor by LabVIEW program for the following three days. The total amount of time spent on the wheel and the total distance run were calculated for each animal.

*Open field test*. Open field test was performed as previously described[13] with minor modifications. The test was performed in a $40 \times 40 \times 40$ cm arena made of white Plexiglas plastic, which was illuminated from above (light intensity of 100 lux on the floor). Mice were placed individually in the center of the area and allowed to freely explore it for 10 min while being recorded. The horizontal locomotor activity was analyzed using the Mouse Behavioral Analysis Toolbox (MouBeAT)[96] for ImageJ. The vertical activity (rearing), defined as a mouse rising on its hind legs, was scored manually.

**In situ hybridization**. Male mouse brains were embedded in 3% low melting agarose (Topvision, R0801) and 50 μm vibratome sections were mounted on Superfrost Plus slides (Thermo Scientific, J1800AMNZ). The sections were fixed for 15 min in 10% formaldehyde, washed in PBS and treated with 10 μg/ml proteinase K in TE (10 mM Tris-HCl, 1 mM EDTA, pH 8.0) for 5 min. The sections were fixed again in 10% formaldehyde for 10 min and washed in PBS, then 0.2 M HCl was added to the slides for 10 min, followed by a PBS wash. The sections were then pre-incubated in 0.1 M triethanolamine HCl at pH 8.0 for 1 min, incubated in 0.1 M triethanolamine HCl with acetic anhydride for 10 min, and then washed in PBS. Probes were denatured for 7 min at 70 °C and diluted at 100 nM in the hybridization buffer: 10 mM NaCl, 5 mM $NaH_2PO_4·H_2O$, 5 mM $Na_2HPO_4·2H_2O$, 5 mM EDTA, 50% formamide, 10% dextran sulfate, 1 μg/μl tRNA, 1 × Denhardt's solution (Eppendorf, 0032007.155). Hybridization was performed with a double DIG-labeled locked nucleic acid (LNA) probe for *Nato3* (/5DiGN/ACTCAGCGTC-TATCTCACCGA/3DiG_N/) (Qiagen) and incubated overnight at 62 °C. The sections were washed twice for 30 min at 62 °C and once at room temperature with 1 x SSC (150 mM NaCl, 15 mM Sodium Citrate, pH 7.0) supplemented with 50% formamide and 0.1% Tween20. Slides were preincubated in 2% blocking reagent (Roche, 11096176001) in MABT (100 mM Maleic Acid, 150 mM NaCl, 0.1% Tween 20) for 20 min followed by 1 h incubation with alkaline phosphatase-anti-digoxigenin antibody (1:500, Roche, 1093274). Alkaline phosphatase activity was detected by incubating slides with Fast Red substrate (DAKO, K0597) for 30 min.

**Microdialysis and quantification of dopamine**. The extracellular dopamine levels in the striatum were measured using microdialysis following the established method[97]. Briefly, 16- to 18-month-old male mice were anaesthetized with iso-flurane using a controlled vaporizer. The animal was placed in a stereotaxic frame and the guide cannula for the microdialysis probe was implanted in the striatum. During the intervention, the animal was covered with a heating pad to keep its body temperature at 37 °C. An analgesic was added to its water (Dafalgan, 2 mg/ml) until the following day. 24 h after the surgery, the microdialysis probe was inserted into the guide cannula, and ringer solution (125 mM NaCl, 2.5 mM KCl, 1.26 mM $CaCl_2$, 1.18 mM $MgCl_2$, 0.20 mM $NaH_2PO_4$) was perfused through the microdialysis probe at a flow at 1.0 μl/min using a high precision pump. The experiments were performed during the light period and the mice were tested in their home cage. After at least a 2 h equilibration period, the dialysates were collected every 30 min for 2.5 h (5 samples in total). Each mouse was analyzed once.

Dopamine levels in the dialysates were quantified using Dopamine Research Elisa assay kit (LDN Labor Diagnostika Nord GmbH & Co. KG, BA E-5300R), according to the manufacturer's instructions. The absorbance at 450 nm was measured by a microplate reader (Cytation 3, BioTek, Winooski, VT, USA). The dopamine concentrations were determined using a standard curve.

**Statistical analysis**. Statistical comparisons were performed using GraphPad Prism software (San Diego, CA). The statistical significance of differences between two groups was calculated using a nonparametric two-tailed Mann–Whitney test. To compare three or more groups, ANOVA followed by Tukey's multiple comparisons test was used. For the circularity analysis, to compare the distribution of mitochondria particles between two or three genotypes, the Kolmogorov–Smirnov test and the Kruskal–Wallis test followed by Dunn's test were applied, respectively. Differences were considered significant at $*p < 0.05$, $**p < 0.01$, $***p < 0.001$, $****p < 0.0001$. ns indicates not significant. Pairwise comparisons are shown as brackets. Only where the differences are not significant and adding brackets reduces the legibility of the plots, brackets and "ns" are omitted. Box plots are used for depicting the group data comprised of fewer than 200 data points. Box boundaries are the 25th and 75th percentiles, the horizontal line across the box is the median, and the whiskers indicate the minimum and maximum values. The group data that consist of 3 data points are shown as scatter dot plots, with the horizontal line across the plot representing the mean. The group data that consist of 200 or more data points are shown as violin plots. The horizontal line across the violin plot is the median and the horizontal dotted lines represent the 25th and 75th percentiles.

**Reporting summary**. Further information on research design is available in the Nature Research Reporting Summary linked to this article.

## Data availability

Source Data are provided with this paper. The datasets generated during the current study are available in the Gene Expression Omnibus (GEO) repository. Accession

numbers and the web links are as follows. ChIP-seq data: GSE156892. whole-head RNA-seq data: GSE156890. PAM neuron RNA-seq data: GSE157589. GTEx portal v7 [https://gtexportal.org/home/]. STRING v10.5 [https://string-db.org] Source data are provided with this paper.

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

## Acknowledgements

We thank Ernst Hafen, Chi-Hon Lee, Junhai Han, the Bloomington *Drosophila* Stock Center, the Vienna Drosophila Resource Center and the Kyoto Stock Center for fly stocks, J-C. Martinou for the *DAT-Cre* mouse strain. We thank the Genomics Platform of iGE3 for ChIP-seq and RNA-seq experiments, the animal facility of the University of Geneva for maintenance of mice colonies, and the Pole Facultaire de Microscopie Electronique (Centre Médical Universitaire, Geneva) for electron microscopy. We are grateful to J-C. Martinou and our lab members for valuable discussions of this work. This research was supported by grants to E.N. from the European Research Council (ERC-StG-311194), the Swiss National Science Foundation (31003A_130387), the Georges and Antoine Claraz Foundation, the Novartis Foundation for Medical-Biomedical Research (19A025), and the Société Académique de Genève. D.T. was partly supported by the Plan Stratégique Sciences Vie (PSVIE) of the University of Geneva. E.V. was partly supported by the Genomics Platform of iGE3.

## Author contributions

E.N. conceptualized the study. E.N., F.M., L.S., D.T. and N.L. designed experiments. F.M., D.T., L.S., N.L., I.N., M.D., E.V., F.P. and P.B.D. performed experiments and analyzed data. F.M., D.T., L.S., M.D., and E.N. produced visualizations of the results and made figures. F.M. and E.N. wrote the manuscript.

## Competing interests

The authors declare no competing interests.
