## [Peer Review File · Nature Communications]

Maintenance of mitochondrial integrity in midbrain dopaminergic neurons governed by a conserved developmental transcription factorREVIEWER COMMENTS

Reviewer #1 (Remarks to the Author):

In this manuscript the authors address the role of Fer2/Nato3 in maintaining mitochondrial integrity in both *Drosophila* and mouse midbrain dopamine neurons (mDA). The research group has previously addressed the role of Fer2 in *Drosophila* in two papers. However, the new *Drosophila* data presented in the current manuscript combined with the mouse analysis has the potential to increase our understanding of how transcription factors serve to protect mDA-neurons from cellular stress. The manuscript is clearly written, and the experiments are in general well suited to address the scientific questions posed. However, there are several issues that must be addressed in order to strengthen the conclusions of the manuscript. In particular, if the cause of the observed locomotor deficits is caused by the loss of mitochondrial integrity needs to be addressed.

Major points

1. In Fig. 1a-b the authors claim that there is significant loss of TH⁺-mDA neurons in the PAM>Lrrk WT and PAM>Lrrk I1915T and that this effect is rescued by Fer2 overexpression. The immunofluorescence images in 1a correspond poorly with the quantification in 1b. Firstly, the three bottom panels in 1a exhibit much higher background compared to the PAM>Lrrk WT and PAM>Lrrk I1915T panels above. This makes proper comparison difficult. Secondly, only the PAM-panel have a clearly higher number of TH⁺ cells than the PAM>Lrrk WT and PAM>Lrrk I1915T panels. It rather looks like the cells are distributed differently with the cells in the upper panels concentrated in a more limited area. DAPI-staining of nuclei must be added to help to determine cell numbers in a better way. The TH-staining must be represented by confocal images with the same degree of background and contrast.
2. Given the ambiguity in the immunofluorescent panel Fig. 1a, the authors must add immunofluorescence (TH/DAPI) panels for the data presented in Fig. 1c-d and Fig. 2d.
3. The reduction of TH⁺-cells is not properly explained. Is it cell death or just a reduction in TH levels that is the cause of the reduction in TH⁺-mDA neurons? This must be addressed.
4. The claim on page 12 that “Our finding that ftz-f1 is necessary for PAM neuron maintenance (Fig. 4f) supports this prediction.” is a substantial overinterpretation of the data presented where OE expression of ftz-f1 hardly change the number of neurons. Statistical significance should not be confused with biological significance. This needs to be toned down in the text.
5. For the secondary target analysis of the 132 DEGs presented in Fig. 4g should include similar analysis but with up- and downregulated genes separated.
6. Since the mouse analysis depends on that Nato3 really is expressed in the mDA-neurons the immunofluorescence and qPCR data in Fig. 5 should be complemented by in situ hybridization.
7. An additional necessary control for NATO3 antibody specificity, NATO3 protein levels and efficiency/specificity of the DatCreNato3-mutant is to perform immunostaining on SN and VTA of mutant mice. This validation is absolutely necessary.

8. The authors conclude from the behavioural tests and the integrity of the nigrostriatal pathways that the locomotor deficits in the mutants likely originate from functional impairments of DA neurons rather than an overt loss of cellular integrity. They show that TH levels are intact in both the midbrain and in the striatal target area. The question then is what is the cause of the motor deficits? Is there a failure to produce dopamine despite normal levels of TH? Is there a loss of dopamine signalling from the mDA-neurons over the synapses to the striatal neurons? This must be addressed at least in writing; how does the loss of mitochondrial integrity affect the function when there still is plenty of TH in the target area. HPLC to measure dopamine synthesis metabolites in the striatum would answer if there is failure in dopamine synthesis. Pre- and postsynaptic e-phys recordings in midbrain/striatum would address if functional signalling properties of the system are affected.

9. To substantiate if the loss of mitochondrial integrity presented in Fig. 7 is a primary cause of the locomotor deficits the authors must examine the earliest time point when mitochondrial integrity is compromised. Since the first signs of locomotor deficits occurs at 13 months (Fig. 5c) loss of mitochondrial integrity after this would imply that the behavioural phenotype depends on an alternative cause. If integrity is compromised much earlier than 13 months when there are no signs of locomotor deficit, it is likely that additional later mechanisms (potentially additive to the effects on the mitochondria) are involved in generating the behavioural phenotype.

Minor points:

1. Are the p-values in Fig. 4a, b & e the adjusted p-values?
2. In the manuscript there is a shift in the reference to the panels of Fig. 6c-h.

Reviewer #2 (Remarks to the Author):

This manuscript reports a transcriptional network controlled by the transcription factor Fer2. Fer2, which is abundantly expressed in the *Drosophila* PAM dopaminergic neurons, is upregulated with age and regulates a variety of genes including mitochondrial genes. Among mitochondrial genes, the overexpression of SOD2 partially suppress the loss of PAM neurons in Fer2 loss-of-function flies. In the latter part, the study characterized mice in which Nato3, a functional ortholog of Fer2, was inactivated in the midbrain dopaminergic neurons after differentiation. While the survival of dopaminergic neurons was not affected, motor functions were impaired in the aged mice. Mitochondrial structure of the dopaminergic neurons was also disorganized in the Nato3 mutant mice.

The authors thoroughly determined direct and indirect targets of Fer2 in *Drosophila*, which is core data in this manuscript. In the meantime, the claim that *Nato3* is a functional homolog of Fer2 is not well-grounded and there are some weak or unaddressed points in this manuscript as described below:

The mitochondrial degeneration by Fer2 loss are analyzed by the morphology using mitoGFP in Fig 1e, f. However, the morphology is relatively unclear compared with data reported in their previous study (Bou Dib, PLoS 2014). In addition, different modalities to estimate mitochondrial degeneration would be required in both fly and mouse studies.

Surprisingly, most direct targets detected in this study is downregulated by Fer2. Fer2 appears to negatively control a transcriptional regulatory network including *ftz-f1*. Thus, the description on page 12 “Our finding that *ftz-f1* is necessary for PAM neuron maintenance (Fig. 4f) supports this prediction” is not appropriate.

SOD2 overexpression partially rescued the loss of PAM neurons (Fig 4h). The authors should determine whether downregulation of SOD2 is the major cause of PAM neurodegeneration by Fer2 loss.

Although the authors claim that *Nato3* is a conserved ortholog of Fer2, there are no data on “conserved targets” in mice. Is SOD2 a possible target to protect mitochondrial degeneration in mammals?

There are several mitochondrial genes that are indirectly regulated by Fer2. To what extent do these contribute to the mitochondrial protection?

Data in Fig 2 appear to be basically similar to those in their previous study (Bou Dib, PLoS 2014) and should be moved to supplemental data.

The description “We propose that *Nato3* cKO mice can serve as a relevant model for prodromal PD, presenting moderate motor deficits and quasi-normal mDA neurons that accumulate abnormal mitochondria”. in Discussion (on page 19) should be more conservative because there is no clinical evidence in the current study. Conversely, data on genetic evidence or altered *Nato3* functions in PD cases would reinforce this manuscript.

Other comments.

In Fig 1b and e, the UAS copy number should be adjusted due to GAL4 titration issue.

Because this manuscript minimally cites references in the last sentences, it is difficult to follow the evidence. For instance, the paragraph in Introduction (on page 9) “The notion that developmental transcription factors play critical.... Fer2 expression persists into adulthood in PAM neurons and increases in response to oxidative stress.” needs appropriate reference citations in each sentence.

In Results section (on page 9), “The binding of transcription factors to chromatin does not necessarily lead to transcriptional regulation of nearby genes”. needs appropriate reference citations. binding > binding.

In Discussion section, the sentence “Importantly, human Nato3 is expressed almost exclusively in the SN among adult tissues”. needs appropriate reference citations.

In Results section (on page 11), “The majority of the transgenic lines used in this assay have been validated in previous studies” needs appropriate reference citations. For RNAi experiments, the authors should determine the knockdown efficiency using multiple RNAi lines if not determined by the previous studies.

Anti-Nato3 immunosignals is preferentially localized in the cytoplasm but not the nuclei (Fig. 5d). It would be better to check specificity of the antibody used here.

Alignment and phylogenetic tree analysis of FER2, FER3, P48, NATO3, TCF15 and TWIST1 would help to understand the evolutionary conservation.

In Fig 5a, data for the other three homolog genes should be shown.

The citations (on page 15), “number of DA neuron cell bodies in the SN and VTA (Fig. 6f-g), as well as the density of TH+ fibers in the striatum (Fig. 6h-i), is wrong. Fig 6e, f and Fig 6g, h are correct.

REPLY to REVIEWER COMMENTS

Reviewer #1 (Remarks to the Author):

In this manuscript the authors address the role of Fer2/Nato3 in maintaining mitochondrial integrity in both Drosophila and mouse midbrain dopamine neurons (mDA). The research group has previously addressed the role of Fer2 in Drosophila in two papers. However, the new Drosophila data presented in the current manuscript combined with the mouse analysis has the potential to increase our understanding of how transcription factors serve to protect mDA-neurons from cellular stress. The manuscript is clearly written, and the experiments are in general well suited to address the scientific questions posed. However, there are several issues that must be addressed in order to strengthen the conclusions of the manuscript. In particular, if the cause of the observed locomotor deficits is caused by the loss of mitochondrial integrity needs to be addressed.

We thank the reviewer for the constructive comments on our work. We address all the concerns in this letter and in the revised manuscript. For your perusal, a marked-up copy of the revised manuscript, showing the changes introduced, is uploaded together with a clean copy.

Major points

1. In Fig. 1a-b the authors claim that there is significant loss of TH+-mDA neurons in the PAM>Lrrk WT and PAM>Lrrk I1915T and that this effect is rescued by Fer2 overexpression. The immunofluorescence images in 1a correspond poorly with the quantification in 1b. Firstly, the three bottom panels in 1a exhibit much higher background compared to the PAM>Lrrk WT and PAM>Lrrk I1915T panels above. This makes proper comparison difficult. Secondly, only the PAM-panel have a clearly higher number of TH+ cells than the PAM>Lrrk WT and PAM>Lrrk I1915T panels. It rather looks like the cells are distributed differently with the cells in the upper panels concentrated in a more limited area. DAPI-staining of nuclei must be added to help to determine cell numbers in a better way. The TH-staining must be represented by confocal images with the same degree of background and contrast.

Thank you for pointing out the poor quality of the images of Fig. 1a. Since PAM neurons are densely packed, it is not easy to display good quality images. However, we made our effort to improve the image quality and replaced the one in Fig. 1a. DAPI staining does not improve legibility of the figure, because many non-TH-cells are in the PAM area and overlapping DAPI signals actually hinder identifying individual TH-positive cells. In contrast, anti-TH staining signal is localized in the cytoplasm, which negatively mark the nucleus. Therefore, it is most straightforward to count TH-positive cells using anti-TH staining.

Please also note that PAM neurons are highly heterogeneous and distributed across many confocal optical planes. If we display maximum z-projection image composed from the entire z-stack including all the PAM neurons, it is simply impossible to distinguish individual cells. Therefore, in the Fig 1a. we display the z-projection images composed from part of the optical planes including PAM neurons. Therefore, the pictures may not appear exactly the same as the data in the plots. However, to quantify PAM neurons, we manually count TH-positive cells through all Z-stacks (not from the z-projection images).

2. Given the ambiguity in the immunofluorescent panel Fig. 1a, the authors must add immunofluorescence (TH/DAPI) panels for the data presented in Fig. 1c-d and Fig. 2d.

3. The reduction of TH+ cells is not properly explained. Is it cell death or just a reduction in TH levels that is the cause of the reduction in TH+-mDA neurons? This must be addressed.

As stated in the answer to comment 1, since DAPI is not a convenient marker for counting PAM neurons, we expressed Red Stinger (nuclear RFP) in PAM neurons to distinguish reduction in TH levels and cell loss. As shown in the Fig. 1c, d, this experiment confirmed the loss of PAM neurons in the flies with *Lrrk* overexpression. *Fer2* overexpression partially rescued PAM neuron loss in mutant *Lrrk* overexpression, but not the wild-type *Lrrk*.

Importantly, the new experiments using RedStinger found that *park¹* mutation does not decrease the number of PAM neurons but only reduces TH levels (Supplementary Fig. 1c, d). Nevertheless, *Fer2* overexpression improves mitochondrial morphology (Fig. 1e, f) and function (Fig. 1g) in PAM neurons in *park¹* flies. These results clearly indicate the role of *Fer2* in maintaining mitochondrial function and morphology. We thank the reviewer for the insightful suggestion, with which we could strengthen our findings.

Please also note that *R58E02-GAL4* is expressed in approximately 80 PAM neurons, whereas approximately 110 PAM neurons can be detected with anti-TH-staining. Therefore, the absolute cell counts are different between Fig. 1b and 1d, but the relative changes in the cell number remains constant, comparing control and *Lrrk* gain-of-function flies.

4. The claim on page 12 that “Our finding that ftz-f1 is necessary for PAM neuron maintenance (Fig. 4f) supports this prediction.” is a substantial overinterpretation of the data presented where OE expression of ftz-f1 hardly change the number of neurons. Statistical significance should not be confused with biological significance. This needs to be toned down in the text.

We agreed and softened the statement (line 303).

5. For the secondary target analysis of the 132 DEGs presented in Fig. 4g should include similar analysis but with up- and downregulated genes separated.

We agreed and we updated the figure (Fig. 4g).

6. Since the mouse analysis depends on that Nato3 really is expressed in the mDA-neurons the immunofluorescence and qPCR data in Fig. 5 should be complemented by in situ hybridization.

7. An additional necessary control for NATO3 antibody specificity, NATO3 protein levels and efficiency/specificity of the DatCreNato3-mutant is to perform immunostaining on SN and VTA of mutant mice. This validation is absolutely necessary.

To address the points 6 and 7, we conducted *in situ* hybridization of *Nato3* together with anti-TH staining in control and *Nato3* cKO mice. We display the results in the revised Fig. 5e and Fig. 6c. *in situ* hybridization experiments validated the expression of *Nato3* mRNA in the SN and VTA, and that cKO effectively reduces *Nato3* expression within DA neurons. While repeating anti-NATO3 immunostaining, we realized that this antibody had a very high background signals and expression differences between control and *Nato3* cKO was obscured. Therefore, we entirely removed anti-NATO3 data in the revised manuscript. We again thank the reviewer for this important suggestion.

8. The authors conclude from the behavioural tests and the integrity of the nigrostriatal pathways that the locomotor deficits in the mutants likely originate from functional impairments of DA neurons rather than an overt loss of cellular integrity. They show that TH levels are intact in both the midbrain and in the striatal target area. The question then is what is the cause of the motor deficits? Is there a failure to produce dopamine despite normal levels of TH? Is there a loss of dopamine signalling from the mDA-neurons over the synapses to the striatal neurons? This must be addressed at least in writing; how does the loss of mitochondrial integrity affect the function when there still is plenty of TH in the target area. HPLC to measure dopamine synthesis metabolites in the striatum would answer if there is failure in dopamine synthesis. Pre- and postsynaptic e-phys recordings in midbrain/striatum would address if functional signalling properties of the system are affected.

To address this well-taken point, we measured striatal extracellular DA concentration using *in vivo* microdialysis in 16-to 18-month-old control and *Nato3* cKO mice. We found no statistically significant differences in DA concentration between genotypes (Fig. 6k). This finding excludes the reduction of DA release as a cause of locomotor impairments in *Nato3* cKO mice.

We acknowledge that we still do not know the direct cause of locomotor deficits, and understanding how mitochondrial defects in mDA neurons in *Nato3* cKO lead to locomotor impairments requires further work in future studies. However, we would also like to emphasize that *PINK1* and *Parkin* mutant mice, the well-established PD mice models that exhibit pronounced mitochondrial defects (Stevens et al., PNAS, 2015; Gautier et al., PNAS, 2018), do display locomotor impairments but do not show loss of mDA neurons or reduction in DA levels in the striatum (Kitada et al. PNAS 2007, Pinto et al., J of Neurosci, 2018, Itier et al. HMG 2003). Changes in the substantia nigra DA neuron excitability and firing pattern is thought to be a part of the cause of the locomotor defects in *PINK1* mutant mice (Bishop et al. J Neurophysiol 2010). Furthermore, iPSC-derived midbrain neurons from *parkin* patients show alterations in the amplitude of spontaneous excitatory postsynaptic currents (sEPSCs) (Zhong et al., Cell Reports, 2017). We hypothesize that *Nato3* cKO mice mDA neurons may similarly have changes in electrophysiological activity that lead to motor impairments. We mention these possibilities in the revised manuscript (line 371 in Results and line 519-526 in discussion).

9. To substantiate if the loss of mitochondrial integrity presented in Fig. 7 is a primary cause of the locomotor deficits the authors must examine the earliest time point when mitochondrial integrity is compromised. Since the first signs of locomotor deficits occurs at 13 months (Fig. 5c) loss of mitochondrial integrity after this would imply that the behavioural phenotype depends on an alternative cause. If integrity is compromised much earlier than 13 months when there are no signs of locomotor deficit, it is likely that additional later mechanisms (potentially additive to the effects on the mitochondria) are involved in generating the behavioural phenotype.

Following your advice, we have analyzed mitochondria in mDA neurons in 6-month-old mice with electron microscopy. We found no difference between *Nato3* cKO and control mice (Fig. 7 d, e). These additional data indicate that mitochondrial defects occur age-dependently as a consequence of *Nato3* ablation.

Minor points:

1. Are the p-values in Fig. 4a, b & e the adjusted p-values?

These are non-adjusted p-values.

2. In the manuscript there is a shift in the reference to the panels of Fig. 6c-h.

Thank you. The mistakes are corrected in the revised manuscript.

Reviewer #2 (Remarks to the Author):

This manuscript reports a transcriptional network controlled by the transcription factor Fer2. Fer2, which is abundantly expressed in the Drosophila PAM dopaminergic neurons, is upregulated with age and regulates a variety of genes including mitochondrial genes. Among mitochondrial genes, the overexpression of SOD2 partially suppress the loss of PAM neurons in Fer2 loss-of-function flies. In the latter part, the study characterized mice in which Nato3, a functional ortholog of Fer2, was inactivated in the midbrain dopaminergic neurons after differentiation. While the survival of dopaminergic neurons was not affected, motor functions were impaired in the aged mice. Mitochondrial structure of the dopaminergic neurons was also disorganized in the Nato3 mutant mice.

The authors thoroughly determined direct and indirect targets of Fer2 in Drosophila, which is core data in this manuscript. In the meantime, the claim that Nato3 is a functional homolog of Fer2 is not well-grounded and there are some weak or unaddressed points in this manuscript as described below:

Thank you for the thorough and constructive assessment of our paper. We address all the concerns in this letter and in the revised manuscript. Please also look at a marked-up copy of the revised manuscript, in which the changes introduced are highlighted.

The mitochondrial degeneration by Fer2 loss are analyzed by the morphology using mitoGFP in Fig 1e, f. However, the morphology is relatively unclear compared with data reported in their previous study (Bou Dib, PLoS 2014). In addition, different modalities to estimate mitochondrial degeneration would be required in both fly and mouse studies.

2.1. Microscope images in the previous manuscript may have been suboptimal. We replaced them to better quality pictures in the revised manuscript (Fig. 1e). Please also note that we improved the images for display, but the quantification was performed from the original images, which show statistically significant differences between genotypes (Fig. 1f).

Furthermore, as suggested, we have monitored mitochondrial membrane potential using TMRM (Tetramethylrhodamine methyl ester) dye. We found that *park1* mutation reduces mitochondrial membrane potential, which is prevented by *Fer2* overexpression. These new data provide evidence that *Fer2* overexpression restore not only morphological abnormality but also functional impairments in mitochondria caused by *parkin* loss of function (Fig. 1g).

Surprisingly, most direct targets detected in this study is downregulated by Fer2. Fer2 appears to negatively control a transcriptional regulatory network including ftz-f1. Thus, the description on page 12 "Our finding that ftz-f1 is necessary for PAM neuron maintenance (Fig. 4f) supports this prediction" is not appropriate.

2.2. Agreed. We corrected the statement in the revised manuscript (line 303).

SOD2 overexpression partially rescued the loss of PAM neurons (Fig 4h). The authors should determine whether downregulation of SOD2 is the major cause of PAM neurodegeneration by Fer2 loss.

2.3. SOD2 is one of many *Fer2* indirect targets. *Fer2* also controls expression of many other genes involved in mitochondrial structure and function, as shown in Fig. 4g. Therefore, we do not think SOD2 downregulation is the major cause of PAM neuron loss, but one of the many factors. The result that SOD2 overexpression partially but not completely rescues PAM neuron loss (Fig. 4h) is congruent with this interpretation.

*Although the authors claim that *Nato3* is a conserved ortholog of *Fer2*, there are no data on “conserved targets” in mice. Is SOD2 a possible target to protect mitochondrial degeneration in mammals?*

2.4. To address this question, we performed anti-SOD2 immunostaining in mDA neurons. We found a significant reduction of SOD2 signal in *Nato3* cKO mice compared to controls (Fig. 7f, g). This new finding suggests that indeed *Fer2* and *Nato3* may have a set of common downstream targets, which control mitochondrial maintenance during aging. However, dissecting the overall genetic program controlled by NATO3 in postmitotic mDA neurons requires further work and is beyond the scope of this manuscript.

*There are several mitochondrial genes that are indirectly regulated by *Fer2*. To what extent do these contribute to the mitochondrial protection?*

2.5. Determining to what extent each of these mitochondrial genes contribute to mitochondrial protection is beyond the scope of this manuscript. However, we would like to emphasize that many mitochondrial genes are regulated by a single transcription factor, which seems to be an efficient way to maintain healthy mitochondria during aging.

Data in Fig 2 appear to be basically similar to those in their previous study (Bou Dib, PLoS 2014) and should be moved to supplemental data.

2.6. In Bou Dib et al. paper, we have used H₂O₂ treatment but the effect on PAM neurons was examined immediately after the treatment. On the contrary, in the current paper, we analyzed PAM neurons 7 days after the H₂O₂ treatment. We added a phrase to clearly distinguish these two experimental conditions in the revised manuscript (line 184-185).

*The description “We propose that *Nato3* cKO mice can serve as a relevant model for prodromal PD, presenting moderate motor deficits and quasi-normal mDA neurons that accumulate abnormal mitochondria”. in Discussion (on page 19) should be more conservative because there is no clinical evidence in the current study. Conversely, data on genetic evidence or altered *Nato3* functions in PD cases would reinforce this manuscript.*

2.7. We agreed and softened the statement (line 517).

A recent study from the group of Ernest Arenas (Toledo et al. Cell Reports 2020) has shown that *Nato3* (*Ferd3l*) is a potential transcriptional target of SREBP1. SREBP1 also upregulates *Foxa2*, which in turn upregulates *Nato3* expression during development (Metzakopian et al. Development 2012). Since *Nato3* is also known to increase *Foxa2* and *Lmx1b* levels (Peterson et al. Neuroscience 2019), these findings altogether suggest that *Nato3* is part of the gene regulatory feedback loop involving SREBP1, *Foxa2*, and *Lmx1b*. Importantly, a polymorphism in the *Srebf1* gene encoding SREBP1 is linked to PD (Do et al. PLOS Genet. 2011; Shulman et al. JAMA Neurol. 2014). Therefore, there is growing evidence that gene regulatory programs involving DA developmental genes, including *Nato3*, have important roles in the survival of mDA neurons during aging and offer an opportunity to study PD and its therapeutic strategies. We include the discussion on the relationship between *Srebf1* and *Nato3* in the revised manuscript (line 464-472).

Other comments.

In Fig 1b and e, the UAS copy number should be adjusted due to GAL4 titration issue.

2.8. In the revised manuscript, we include the data, in which UAS-RedStinger, UAS-Lrrk WT or mutant, and UAS-Fer2 are co-expressed with the same driver (Fig. 1c, d). The data show that despite the two genotypes (PAM > UAS-RedStinger, UAS-Lrrk WT, UAS-Fer2 and PAM > UAS-Red Stinger, UAS-Lrrk I1915T, UAS-Fer2) express the same number of UAS transgenes, they exhibit a different number of RedStinger-positive cell (comparing these two groups, *p<0.05 by one-way ANOVA followed by a Turkey's test for multiple group comparison. This statistical comparison is not indicated in the Fig. 1d to avoid overcrowding the plot). These results suggest that the rescue of the cell count number is likely not due to the titration of GAL4.

Because this manuscript minimally cites references in the last sentences, it is difficult to follow the evidence. For instance, the paragraph in Introduction (on page 9) "The notion that developmental transcription factors play critical.... Fer2 expression persists into adulthood in PAM neurons and increases in response to oxidative stress." needs appropriate reference citations in each sentence.

In Results section (on page 9), "The biding of transcription factors to chromatin does not necessarily lead to transcriptional regulation of nearby genes". needs appropriate reference citations. biding > binding.

2.9. Thank you for the advice. Mistakes were corrected in the revised manuscript. We also added missing citations (lines 88 and 217).

In Discussion section, the sentence "Importantly, human Nato3 is expressed almost exclusively in the SN among adult tissues". needs appropriate reference citations.

2.10. This sentence refers to the *NATO3* mRNA expression profile in human RNA-seq data from the GTEx consortium, as displayed in Fig. 5b. We added the necessary reference (line 456).

In Results section (on page 11), “The majority of the transgenic lines used in this assay have been validated in previous studies” needs appropriate reference citations. For RNAi experiments, the authors should determine the knockdown efficiency using multiple RNAi lines if not determined by the previous studies.

2.11. Majority of FER2 direct targets are downregulated by FER2, therefore most of the genotypes analyzed in our screen consist in an overexpression of the target gene. In the Supplementary Data 3, we include the reference citations of the fly lines used in the mini-screen.

As suggested, we also tested the knockdown efficiency of a few RNAi lines used in the assay, by expressing the RNAi with a ubiquitous Tubulin driver and assessing the levels of the targeted gene by RT-qPCR. We found that two lines (*Theg* RNAi and *CG1578* RNAi) had low knockdown efficiency (~ 30% reduction). We removed the data generated using these two lines, as we could not determine whether or not the lack of the effect on PAM neurons was due to low RNAi efficiency. Tubulin-driven and Actin-driven expression of *CG4998* RNAi resulted in lethality, suggesting that *CG4998* RNAi efficiently knocks down *CG4998* expression. We added a statement regarding the RNAi efficiency in the revised manuscript (line 261-265).

Anti-Nato3 immunosignals is preferentially localized in the cytoplasm but not the nuclei (Fig. 5d). It would be better to check specificity of the antibody used here.

2.12. To address this important point, we tested anti-NATO3 antibodies in *Nato3 cKO* mice, and found that they have high background staining that obscures the true NATO3 signal. Therefore, we entirely removed the data that used anti-NATO3 staining in the revised manuscript. Instead, we newly conducted *in situ* hybridization of *Nato3* together with anti-TH staining in control and *Nato3 cKO* mice. We display the results in the revised Fig. 5e and Fig. 6c, d. *in situ* hybridization experiments validated the expression of *Nato3* mRNA in the SN and VTA, and show that cKO effectively reduces *Nato3* expression within DA neurons. We thank the reviewer for the important suggestion.

Alignment and phylogenetic tree analysis of FER2, FER3, P48, NATO3, TCF15 and TWIST1 would help to understand the evolutionary conservation.

In Fig 5a, data for the other three homolog genes should be shown.

2.13. We added a figure of Phylogenetic tree analysis of FER2 homologs in Fig. 5a to highlight that the closest mammalian homolog of FER2 is NATO3. Additionally, we also display a phylogenetic tree including FER1 and FER3 in Supplementary Fig. 4a. As shown in the Supplementary Fig. 4a, NATO3 sequence is structurally close to FER2 and FER3 among *Fer2* homologs in *Drosophila*. Therefore, we do not exclude that FER3 may have similar function as NATO3, which is an interesting subject for future studies, as mentioned in the Discussion (446-448). We found that protein sequence alignment does not add more information than phylogenetic trees, so decided to display only the phylogenetic trees.

The citations (on page 15), “number of DA neuron cell bodies in the SN and VTA (Fig. 6f-g), as well as the density of TH+ fibers in the striatum (Fig. 6h-i), is wrong. Fig 6e, f and Fig 6g, h are correct.

2.14. Mistakes have been corrected.

REVIEWER COMMENTS

Reviewer #1 (Remarks to the Author):

The authors have addressed my concerns, except for point 9. Their added analysis has not resolved whether loss of mitochondrial integrity precedes locomotor deficits. The new analysis at six months only show that mitochondrial integrity is not affected at this time-point. The question was when it first occurs. If it indeed occurs before motor impairment it strengthens the hypothesis that it is the cause of the motor deficits. Is the mitochondrial integrity perturbed before occurrence of motor deficits? This question must be answered.

Reviewer #2 (Remarks to the Author):

The authors responded appropriately to this reviewer's comments. As a result, the idea of the interspecies conservation between Fer2 and Nato3 in dopaminergic neurons would be strengthened. There are some questions from new additional data and the quality of paper would be improved when the following are addressed.

There is no scale bar in Fig 1c.

The GO term of nodes in Fig 4g should be detailed in supplemental data.

In Fig 5e, the authors show the distribution of Nato3 transcripts and mentioned that the signal was detected even in TH-negative cells. Thus, it would be more valuable to add lower magnification images on Nato3 distribution in the VTA and SNc regions. When the signals are too small to be shown by low magnification, a schematic diagram can be used.

In Fig. 5k, the legend says that data are presented as fold change, but the graph looks like a real value.

In Fig. 7a, there are mitochondrial aggregates even in the TH-negative cells, which appears to be no difference from TH-positive cells. Does this mean that there is no correlation between mitochondrial aggregation shown by the COX4 staining and cristae degeneration shown by the EM analysis? Please provide a supplementary explanation.

In Fig 7f, the authors show a decrease in SOD2 at the protein levels by immunostaining. In contrast, SOD2 is regulated at the transcript levels downstream of Fer2 in flies. Is SOD2 also regulated at the transcript levels by Nato3?

Please provide the number of mice used for mitochondrial quantification in Fig 7d,e.

In Discussion (line 442), the authors mention the similarity of bHLH domains. Thus, it would be better to have an alignment analysis of bHLH domains in Fig S4a.

REVIEWER COMMENTS

Reviewer #1 (Remarks to the Author):

The authors have addressed my concerns, except for point 9. Their added analysis has not resolved whether loss of mitochondrial integrity precedes locomotor deficits. The new analysis at six months only show that mitochondrial integrity is not affected at this time-point. The question was when it first occurs. If it indeed occurs before motor impairment it strengthens the hypothesis that it is the cause of the motor deficits. Is the mitochondrial integrity perturbed before occurrence of motor deficits? This question must be answered.

To address this point, we have analyzed mitochondrial morphology (anti-COX4) and SOD2 levels by immunohistochemistry in the SN DA neurons of 11 months old *Nato3* cKO and control mice. (Immunohistochemistry was used instead of electron microscopy, because the electron microscopy specialist has retired and there is currently no one who can perform the same experiments in our electron microscopy facility.) As shown in Fig 7a, b and 7f, g, the results of these new sets of experiments show that mitochondrial dysmorphism and SOD2 level reduction appear already at 11 months, before the onset of locomotor impairment (13 months). Taken together with the electron microscopy data that the mitochondrial morphology in *Nato3* cKO is normal at 6 months (Fig. 7c-e), these results indicate that mitochondrial impairment occurs age-dependently before the onset of locomotor deficits, and further suggest that mitochondrial dysfunction is likely causally involved in the locomotor impairments (line 396-400). In line with our conclusion, a recent paper (Gonzalez-Rodriguez et al. Nature 2021) demonstrates that conditional disruption of mitochondrial complex I in differentiated DA neurons causes progressive parkinsonism.

Reviewer #2 (Remarks to the Author):

*The authors responded appropriately to this reviewer's comments. As a result, the idea of the interspecies conservation between *Fer2* and *Nato3* in dopaminergic neurons would be strengthened. There are some questions from new additional data and the quality of paper would be improved when the following are addressed.*

There is no scale bar in Fig 1c.

We thank the reviewer for noticing it. We added the scale bar in the revised Fig.1c.

The GO term of nodes in Fig 4g should be detailed in supplemental data.

We added the description of GO terms in Supplementary Data 5.

*In Fig 5e, the authors show the distribution of *Nato3* transcripts and mentioned that the signal was detected even in TH-negative cells. Thus, it would be more valuable to add lower magnification images on *Nato3* distribution in the VTA and SNc regions. When the signals are too small to be shown by low magnification, a schematic diagram can be used.*

We added a lower magnification image of *Nato3 in situ* staining of SN and VTA (Fig. 5e).

In Fig. 5k, the legend says that data are presented as fold change, but the graph looks like a real value.

The graph shows indeed real values, thank you for spotting the mistake. We corrected the mistake in the legend.

In Fig. 7a, there are mitochondrial aggregates even in the TH-negative cells, which appears to be no difference from TH-positive cells. Does this mean that there is no correlation between mitochondrial aggregation shown by the COX4 staining and cristae degeneration shown by the EM analysis? Please provide a supplementary explanation.

The Reviewer raises a good point. Here we have not systematically analyzed mitochondrial morphology of non-TH-positive cells, since they include various neuronal types and glia, and without distinguishing among the different cell types the data will be difficult to interpret. In the EM data (Fig. 7c), we find aberrant mitochondria selectively in DA neurons, at least within the images we have analyzed. Therefore, we tend to think that mitochondrial morphology is primarily affected in DA neurons. However, as Reviewer suggested, there may be a possibility that mitochondria in non-TH-positive cells are affected in a non-cell-autonomous manner by *Nato3* conditional ablation in DA neurons. Whether it is the case, which cell types are affected and why, are indeed interesting questions to address in future studies.

*In Fig 7f, the authors show a decrease in SOD2 at the protein levels by immunostaining. In contrast, SOD2 is regulated at the transcript levels downstream of *Fer2* in flies. Is SOD2 also regulated at the transcript levels by *Nato3*?*

To address this point, we measured *Sod2* mRNA levels by RT-qPCR in the ventral midbrain of 16–18-month-old *Nato3* cKO and control mice (Fig. 7h, and line 406-410). The data show no significant difference between the two genotypes, indicating that *Sod2* is not regulated by *Nato3* at the transcript levels. However, we'd like to emphasize the fact that *Fer2* and *Nato3* both regulate SOD2 levels despite using

different mechanisms highlights their shared role of protecting mitochondrial function. This point is added in the discussion (line 505-506).

Please provide the number of mice used for mitochondrial quantification in Fig 7d,e.

We added the number of mice in the legend.

In Discussion (line 442), the authors mention the similarity of bHLH domains. Thus, it would be better to have an alignment analysis of bHLH domains in Fig S4a.

We added the bHLH domains alignment analysis in Supplementary Figure 4a.

REVIEWERS' COMMENTS

Reviewer #1 (Remarks to the Author):

The differences in particle circularity (7b) and mitochondrial roundness (7e) are homeopathic at best. The reduced levels of SOD2 (7f-g) are more likely to be connected to the phenotype and this must be reflected in the text. Besides that, the authors have addressed my concerns.

Reviewer #2 (Remarks to the Author):

The authors generally responded to the revisions appropriately and I have no further comments although there are still unresolved issues in the analysis of the mice.

The question of whether rare variants of Nato3 is linked to Parkinson's disease and the mechanism of mitochondrial maintenance by Nato3 remain to be addressed in future studies. I look forward to future studies.

REPLY to REVIEWER COMMENTS

Reviewer #1 (Remarks to the Author):

The differences in particle circularity (7b) and mitochondrial roundness (7e) are homeopathic at best. The reduced levels of SOD2 (7f-g) are more likely to be connected to the phenotype and this must be reflected in the text. Besides that, the authors have adressed my concerns.

As suggested, we rephrased the sentences concerning this point in the revised manuscript (lines 378 and 530).

Reviewer #2 (Remarks to the Author):

The authors generally responded to the revisions appropriately and I have no further comments although there are still unresolved issues in the analysis of the mice.

The question of whether rare variants of Nato3 is linked to Parkinson's disease and the mechanism of mitochondrial maintenance by Nato3 remain to be addressed in future studies. I look forward to future studies.